# FAIR NORMALIZING FLOWS

**Mislav Balunović**
ETH Zurich
`mislav.balunovic@inf.ethz.ch`

**Anian Ruoss**[*]
ETH Zurich, DeepMind
`anianr@deepmind.com`

**Martin Vechev**
ETH Zurich
`martin.vechev@inf.ethz.ch`

## ABSTRACT

Fair representation learning is an attractive approach that promises fairness of downstream predictors by encoding sensitive data. Unfortunately, recent work has shown that strong adversarial predictors can still exhibit unfairness by recovering sensitive attributes from these representations. In this work, we present Fair Normalizing Flows (FNF), a new approach offering more rigorous fairness guarantees for learned representations. Specifically, we consider a practical setting where we can estimate the probability density for sensitive groups. The key idea is to model the encoder as a normalizing flow trained to minimize the statistical distance between the latent representations of different groups. The main advantage of FNF is that its exact likelihood computation allows us to obtain guarantees on the maximum unfairness of any potentially adversarial downstream predictor. We experimentally demonstrate the effectiveness of FNF in enforcing various group fairness notions, as well as other attractive properties such as interpretability and transfer learning, on a variety of challenging real-world datasets.

## 1 INTRODUCTION

As machine learning is increasingly being used in scenarios that can negatively affect humans (Brennan et al., 2009; Khandani et al., 2010; Barocas & Selbst, 2016), fair representation learning has become one of the most promising ways to encode data into new, unbiased representations with high utility. Concretely, the goal is to ensure that representations have two properties: (i) they are informative for various prediction tasks of interest, (ii) sensitive attributes of the original data (e.g., race) cannot be recovered from the representations. Perhaps the most prominent approach for learning fair representations is adversarial training (Edwards & Storkey, 2016; Madras et al., 2018; Xie et al., 2017; Song et al., 2019; Roy & Boddeti, 2019), which jointly trains an encoder trying to transform data into a fair representation with an adversary attempting to recover sensitive attributes from the representation. However, several recent lines of work (Feng et al., 2019; Moyer et al., 2018; Elazar & Goldberg, 2018; Xu et al., 2020; Gupta et al., 2021; Song & Shmatikov, 2020) have noticed that these approaches do not produce truly fair representations: stronger adversaries *can* in fact recover sensitive attributes. Clearly, this could allow malicious or ignorant users to use the provided representations to discriminate. This problem emerges at a time when regulators are crafting rules (Whittaker et al., 2018; EU, 2021; FTC, 2021) on the fair usage of AI, stating that any entity that cannot guarantee non-discrimination would be held accountable for the produced data. This raises the question: *Can we learn representations which provably guarantee that sensitive attributes cannot be recovered?*

**This work** Following prior work, we focus on tabular datasets used for tasks such as loan or insurance assessment where fairness is of high relevance. We assume that the original input data $x$ comes from two probability distributions $p_0$ and $p_1$, representing groups with sensitive attributes $a = 0$ and $a = 1$, respectively. In the cases where distributions $p_0$ and $p_1$ are known, we will obtain provable fairness guarantees, and otherwise we perform density estimation and obtain guarantees with respect to the estimated distribution. In our experimental evaluation we confirm that the bounds

---

[*]Work performed while at ETH Zurich.

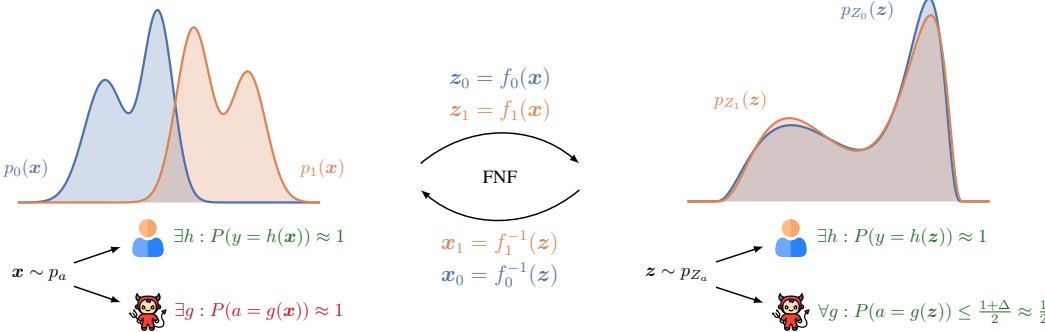

Figure 1: Overview of Fair Normalizing Flows (FNF). There are two encoders, $f_0$ and $f_1$, that transform the two input distributions $p_0$ and $p_1$ into latent distributions $p_{Z_0}$ and $p_{Z_1}$ with a small statistical distance $\Delta \approx 0$. Without FNF, a strong adversary $g$ can easily recover sensitive attribute $a$ from the original input $\boldsymbol{x}$, but once inputs are passed through FNF, we are guaranteed that *any* adversary that tries to guess sensitive attributes from latent $\boldsymbol{z}$ cannot be significantly better than random chance. At the same time, we can ensure that any benevolent user $h$ maintains high utility.

computed on the estimated distribution in practice also bound adversarial accuracy on the true distribution, meaning that density estimation works well for the setting we consider.

To address the above challenges, we propose Fair Normalizing Flows (FNF), a new method for learning fair representations with guarantees. In contrast to other approaches where encoders are standard feed-forward neural networks, we instead model the encoder as a normalizing flow (Rezende & Mohamed, 2015). Fig. 1 provides a high-level overview of FNF. As shown on the left in Fig. 1, using raw inputs $\boldsymbol{x}$ allows us to train high-utility classifiers $h$, but at the same time does not protect against the existence of a malicious adversary $g$ that can predict a sensitive attribute $a$ from the features in $\boldsymbol{x}$. Our architecture consists of two flow-based encoders $f_0$ and $f_1$, where flow $f_a$ transforms probability distribution $p_a(\boldsymbol{x})$ into $p_{Z_a}(\boldsymbol{z})$ by mapping $\boldsymbol{x}$ into $\boldsymbol{z} = f_a(\boldsymbol{x})$. The goal of the training procedure is to minimize the distance $\Delta$ between the resulting distributions $p_{Z_0}(\boldsymbol{z})$ and $p_{Z_1}(\boldsymbol{z})$ so that an adversary cannot distinguish between them. Intuitively, after training our encoder, each latent representation $\boldsymbol{z}$ can be inverted into original inputs $x_0 = f_0^{-1}(\boldsymbol{z})$ and $x_1 = f_1^{-1}(\boldsymbol{z})$ that should ideally have similar probability w.r.t. $p_0$ and $p_1$, meaning that even the optimal adversary cannot distinguish which of them actually produced latent $\boldsymbol{z}$. Crucially, as normalizing flows enable us to compute the exact likelihood in the latent space, for trained encoders we can upper bound the accuracy of *any* adversary with $\frac{1+\Delta}{2}$, which should be small if training was successful. Furthermore, the distance $\Delta$ provides a tight upper bound (Madras et al., 2018) on common fairness notions such as demographic parity (Dwork et al., 2012) and equalized odds (Hardt et al., 2016). As shown on the right in Fig. 1, we can still train high-utility classifiers $h$ using our representations, but now we can actually *guarantee* that no adversary $g$ can recover sensitive attributes better than chance.

We empirically demonstrate that FNF can substantially increase provable fairness without significantly sacrificing accuracy on several common datasets. Additionally, we show that the invertibility of FNF enables algorithmic recourse, allowing us to examine how to reverse a negative decision outcome.

**Main contributions**   Our key contributions are:

- A novel fair representation learning method, called Fair Normalizing Flows (FNF), which guarantees that the sensitive attributes cannot be recovered from the learned representations at the cost of a small decrease in classification accuracy.

- Experimental evaluation demonstrating that FNF can provably remove sensitive attributes from the representations, while keeping accuracy for the prediction task sufficiently high.

- Extensive investigation of algorithmic recourse and applications of FNF to transfer learning.

## 2  RELATED WORK

In this work, we focus on group fairness, which requires certain classification statistics to be equal across different groups of the population. Concretely, we consider demographic parity (Dwork et al., 2012), equalized odds (Hardt et al., 2016), and equality of opportunity (Hardt et al., 2016), which are widely studied in the literature (Edwards & Storkey, 2016; Madras et al., 2018; Zemel et al., 2013). Algorithms enforcing such fairness notions target various stages of the machine learning pipeline: Pre-processing methods transform sensitive data into an unbiased representation (Zemel et al., 2013; McNamara et al., 2019), in-processing methods modify training by incorporating fairness constraints (Kamishima et al., 2011; Zafar et al., 2017), and post-processing methods change the predictions of a pre-trained classifier (Hardt et al., 2016). Here, we consider fair representation learning (Zemel et al., 2013), which computes data representations that hide sensitive information, e.g. group membership, while maintaining utility for downstream tasks and allowing transfer learning.

**Fair representation learning**  Fair representations can be learned with a variety of different approaches, including variational autoencoders (Moyer et al., 2018; Louizos et al., 2016), adversarial training (Edwards & Storkey, 2016; Madras et al., 2018; Xie et al., 2017; Song et al., 2019; Roy & Boddeti, 2019; Liao et al., 2019; Jaiswal et al., 2020; Feng et al., 2019), and disentanglement (Creager et al., 2019; Locatello et al., 2019). Adversarial training methods minimize a lower bound on demographic parity, namely an adversary's accuracy for predicting the sensitive attributes from the latent representation. However, since these methods only empirically evaluate worst-case unfairness, adversaries that are not considered during training can still recover sensitive attributes from the learned representations (Feng et al., 2019; Moyer et al., 2018; Elazar & Goldberg, 2018; Xu et al., 2020; Gupta et al., 2021; Song & Shmatikov, 2020). These findings illustrate the necessity of learning representations with provable guarantees on the maximum recovery of sensitive information regardless of the adversary, which is precisely the goal of our work. Prior work makes first steps in this direction: Gupta et al. (2021) upper bound a monotonically increasing function of demographic parity with the mutual information between the latent representation and sensitive attributes. However, the monotonic nature of this bound prevents computing guarantees on the reconstruction power of the optimal adversary. Feng et al. (2019) minimize the Wasserstein distance between latent distributions of different protected groups, but only provide an upper bound on the performance of any Lipschitz continuous adversary. However, as we will show, the optimal adversary is generally discontinuous. A concurrent work (Cerrato et al., 2022) also learns fair representations using normalizing flows, but different to us, they do not use exact likelihood computation to provide theoretical fairness guarantees.

**Provable fairness guarantees**  The ongoing development of guidelines on the fair usage of AI (Whittaker et al., 2018; EU, 2021; FTC, 2021) has spurred interest in provably fair algorithms. Unlike this work, the majority of these efforts (McNamara et al., 2019; John et al., 2020; Urban et al., 2020; Ruoss et al., 2020) focus on individual fairness. Individual fairness is also tightly linked to differential privacy (Dwork et al., 2012; 2006), which guarantees that an attacker cannot infer whether a given individual was present in the dataset or not, but these models can still admit reconstruction of sensitive attributes by leveraging population-level correlations (Jagielski et al., 2019). Group fairness certification methods (Albarghouthi et al., 2017; Bastani et al., 2019; Segal et al., 2020) generally only focus on certification and, unlike our work, do not learn representations that are provably fair.

## 3  BACKGROUND

We assume that the data $(\boldsymbol{x}, a) \in \mathbb{R}^d \times \mathcal{A}$ comes from a probability distribution $p$, where $\boldsymbol{x}$ represents the features and $a$ represents a sensitive attribute. In this work, we focus on the case where the sensitive attribute is binary, meaning $\mathcal{A} = \{0, 1\}$. Given $p$, we can define the conditional probabilities as $p_0(\boldsymbol{x}) = P(\boldsymbol{x} \mid a = 0)$ and $p_1(\boldsymbol{x}) = P(\boldsymbol{x} \mid a = 1)$. We are interested in classifying each sample $(\boldsymbol{x}, a)$ to a label $y \in \{0, 1\}$, which may or may not be correlated with the sensitive attribute $a$. Our goal is to build a classifier $\hat{y} = h(\boldsymbol{x})$ that tries to predict $y$ from the features $\boldsymbol{x}$, while satisfying certain notions of fairness. Next, we present several definitions of fairness relevant for this work.

**Fairness criteria**  A classifier $h$ satisfies *demographic parity* if it assigns positive outcomes to both sensitive groups equally likely, i.e., $P(h(\boldsymbol{x}) = 1 \mid a = 0) = P(h(\boldsymbol{x}) = 1 \mid a = 1)$. If demographic parity cannot be satisfied, we consider demographic parity distance, defined as

$|\mathbb{E}\left[h(\boldsymbol{x}) \mid a=0\right]-\mathbb{E}\left[h(\boldsymbol{x}) \mid a=1\right]|$. An issue with demographic parity occurs if the base rates differ among the attributes, i.e., $P(y=1 \mid a=0) \neq P(y=1 \mid a=1)$. In that case, even the ground truth label $y$ does not satisfy demographic parity. Thus, Hardt et al. (2016) introduced *equalized odds*, which requires that $P(h(\boldsymbol{x})=1 \mid y=y_0, a=0)=P(h(\boldsymbol{x})=1 \mid y=y_0, a=1)$ for $y_0 \in \{0,1\}$.

**Fair representations** Instead of directly predicting $y$ from $\boldsymbol{x}$, Zemel et al. (2013) introduced the idea of learning *fair representations* of data. The idea is that a data producer preprocesses the original data $\boldsymbol{x}$ to obtain a new representation $\boldsymbol{z}=f(\boldsymbol{x}, a)$. Then, any data consumer, who is using this data to solve a downstream task, can use $\boldsymbol{z}$ as an input to the classifier instead of the original data $\boldsymbol{x}$. Thus, if the data producer can ensure that data representation is fair (w.r.t. some fairness notion), then all classifiers employing this representation will automatically inherit the fairness property. However, due to inherent biases of the dataset, this fairness increase generally results in a small accuracy decrease (see Appendix C for an investigation of this tradeoff in the context of our method).

**Normalizing flows** Flow-based generative models (Rezende & Mohamed, 2015; Dinh et al., 2015; 2016; Kingma & Dhariwal, 2018) provide an attractive framework for transforming any probability distribution $q$ into another distribution $\bar{q}$. Accordingly, they are often used to estimate densities from data using the *change of variables* formula on a sequence of invertible transformations, so-called normalizing flows (Rezende & Mohamed, 2015). In this work, however, we mainly leverage the fact that flow models sample a latent variable $\boldsymbol{z}$ from a density $\bar{q}(\boldsymbol{z})$ and apply an invertible function $f_{\boldsymbol{\theta}}$, parametrized by $\boldsymbol{\theta}$, to obtain datapoint $\boldsymbol{x}=f_{\boldsymbol{\theta}}^{-1}(\boldsymbol{z})$. Given a density $q(\boldsymbol{x})$, the exact log-likelihood is then obtained by applying the change of variables formula $\log q(\boldsymbol{x})=\log \bar{q}(\boldsymbol{z})+\log|\det(d\boldsymbol{z}/d\boldsymbol{x})|$. Thus, for $f_{\boldsymbol{\theta}}=f_1 \circ f_2 \circ \ldots \circ f_K$ with $\boldsymbol{r}_0=\boldsymbol{x}$, $f_i(\boldsymbol{r}_{i-1})=\boldsymbol{r}_i$, and $\boldsymbol{r}_K=\boldsymbol{z}$, we have

$$\log q(\boldsymbol{x})=\log \bar{q}(\boldsymbol{z})+\sum_{i=1}^{K} \log|\det(d\boldsymbol{r}_i/d\boldsymbol{r}_{i-1})|. \tag{1}$$

A clever choice of transformations $f_i$ (Rezende & Mohamed, 2015; Dinh et al., 2015; 2016) makes the computation of the log-determinant tractable, resulting in efficient training and sampling. Alternative generative models cannot compute the exact log-likelihood (e.g., VAEs (Kingma & Welling, 2014), GANs (Goodfellow et al., 2014)) or have inefficient sampling (e.g., autoregressive models). Our approach is also related to discrete flows (Tran et al., 2019; Hoogeboom et al., 2019) and alignment flows (Grover et al., 2020; Usman et al., 2020). However, alignment flows jointly learn the density and the transformation, unlike the fairness setting where these are computed by different entities.

## 4 MOTIVATION

In this section, we motivate our approach by highlighting some key issues with fair representation learning based on adversarial training. Consider a distribution of samples $\boldsymbol{x}=(x_1, x_2) \in \mathbb{R}^2$ divided into two groups, shown as blue and orange in Fig. 2. The first group with a sensitive attribute $a=0$ has a distribution $(x_1, x_2) \sim p_0$, where $p_0$ is a mixture of two Gaussians $\mathcal{N}([-3,3], I)$ and $\mathcal{N}([3,3], I)$. The second group with a sensitive attribute $a=1$ has a distribution $(x_1, x_2) \sim p_1$, where $p_1$ is a mixture of two Gaussians $\mathcal{N}([-3,-3], I)$ and $\mathcal{N}([3,-3], I)$. The label of a point $(x_1, x_2)$ is defined by $y=1$ if $\text{sign}(x_1)=\text{sign}(x_2)$ and $y=0$ otherwise. Our goal is to learn a data representation $\boldsymbol{z}=f(\boldsymbol{x}, a)$ such that it is *impossible* to recover $a$ from $\boldsymbol{z}$, but still possible to predict target $y$ from $\boldsymbol{z}$. Note that such a representation exists for our task: simply setting $\boldsymbol{z}=f(\boldsymbol{x}, a)=(-1)^a \boldsymbol{x}$ makes it impossible to predict whether a particular $\boldsymbol{z}$ corresponds to $a=0$ or $a=1$, while still allowing us to train a classifier $h$ with essentially perfect accuracy (e.g., $h(\boldsymbol{z})=\mathbb{1}_{\{z_1>0\}}$).

**Adversarial training for fair representations** Adversarial training (Edwards & Storkey, 2016; Madras et al., 2018) is an approach that trains encoder $f$ and classifier $h$ jointly with an adversary $g$ trying to predict the sensitive attribute $a$. While the adversary tries to minimize its loss $\mathcal{L}_{adv}$, the encoder $f$ and classifier $h$ are trying to maximize $\mathcal{L}_{adv}$ and minimize the classification loss $\mathcal{L}_{clf}$ as

$$\min_{f, h} \max_{g \in \mathcal{G}} \mathbb{E}_{(\boldsymbol{x}, a) \sim D}\left[\mathcal{L}_{clf}(f(\boldsymbol{x}, a), h)-\gamma \mathcal{L}_{adv}(f(\boldsymbol{x}, a), g)\right], \tag{2}$$

where $\mathcal{G}$ denotes the model family of adversaries, e.g., neural networks, considered during training. Unfortunately, there are two key issues with adversarial training. First, it yields a non-convex

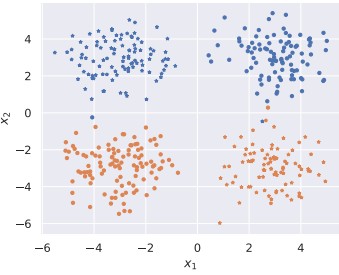

Figure 2: Samples from our example distribution. The blue group ($a = 0$) is sampled from $p_0$ and the orange group ($a = 1$) is sampled from $p_1$.

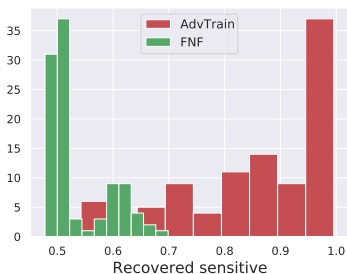

Figure 3: Sensitive attribute recovery rates for adversarial training and fair normalizing flows (FNF) with 100 different random seeds.

optimization problem, which usually cannot be solved to optimality because of saddle points. Second, it assumes that the adversary $g$ comes from a fixed model family $\mathcal{G}$, which means that even if the optimal $g \in \mathcal{G}$ cannot recover the sensitive attribute $a$, adversaries from other model families can still do so as demonstrated in recent work (Feng et al., 2019; Moyer et al., 2018; Elazar & Goldberg, 2018; Xu et al., 2020; Gupta et al., 2021). To investigate these issues, we apply adversarial training to learn representations for our synthetic example, and measure how often the sensitive attributes can be recovered from learned representations. Our results, shown in Fig. 3, repeated 100 times with different seeds, demonstrate that adversarial training is unstable and rarely results in truly fair representations (where only 50% can be recovered). In Section 6 we follow up on recent work and show that several adversarial fair representation learning approaches do not work against adversaries from a different model familiy (e.g., larger networks). In Fig. 3 we show that our approach, introduced next, can reliably produce fair representations without affecting the utility.

## 5 FAIR NORMALIZING FLOWS

Throughout this section we will assume knowledge of prior distributions $p_0(\boldsymbol{x})$ and $p_1(\boldsymbol{x})$. At the end of the section, we discuss the required changes if we only work with estimates. Let $\mathcal{Z}_0$ and $\mathcal{Z}_1$ denote conditional distributions of $z = f(\boldsymbol{x}, a)$ for $a \in \{0, 1\}$, and let $p_{Z_0}$ and $p_{Z_1}$ denote their respective densities. Madras et al. (2018) have shown that bounding the *statistical distance* $\Delta(p_{Z_0}, p_{Z_1})$ between $\mathcal{Z}_0$ and $\mathcal{Z}_1$ provides an upper bound on the unfairness of any classifier $h$ built on top of the representation encoded by $f$. The statistical distance between $\mathcal{Z}_0$ and $\mathcal{Z}_1$ is defined similarly to maximum mean discrepancy (MMD) (Gretton et al., 2006) between the two distributions:

$$\Delta(p_{Z_0}, p_{Z_1}) \triangleq \sup_{\mu \in \mathcal{B}} |\mathbb{E}_{\boldsymbol{z} \sim \mathcal{Z}_0}[\mu(\boldsymbol{z})] - \mathbb{E}_{\boldsymbol{z} \sim \mathcal{Z}_1}[\mu(\boldsymbol{z})]|, \tag{3}$$

where $\mu \colon \mathbb{R}^d \to \{0, 1\}$ is a function in a set of all binary classifiers $\mathcal{B}$ trying to discriminate between $\mathcal{Z}_0$ and $\mathcal{Z}_1$. If we can train an encoder to induce latent distributions $\mathcal{Z}_0$ and $\mathcal{Z}_1$ with statistical distance below some threshold, then we can both upper bound the maximum adversarial accuracy by $(1 + \Delta(p_{Z_0}, p_{Z_1}))/2$ and, using the bounds from Madras et al. (2018), obtain guarantees for demographic parity and equalized odds of any downstream classifier $h$. Such guarantees are unattainble for adversarial training, which minimizes a lower bound of $\Delta(p_{Z_0}, p_{Z_1})$. In contrast, we learn fair representations that allow computing the optimal adversary $\mu^*$ attaining the supremum in Eq. (3) and thus enable exact evaluation of $\Delta(p_{Z_0}, p_{Z_1})$.

**Optimal adversary** In the following lemma we state the form of an optimal adversary which attains the supremum in the definition of statistical distance in Eq. (3). We show the proof in Appendix A.1.

**Lemma 5.1.** *The adversary $\mu^*$ attaining the supremum in the definition of $\Delta(p_{Z_0}, p_{Z_1})$ can be defined as $\mu^*(\boldsymbol{z}) = \mathbb{1}_{\{p_{Z_0}(\boldsymbol{z}) \leq p_{Z_1}(\boldsymbol{z})\}}$, namely it evaluates to 1 if and only if $p_{Z_0}(\boldsymbol{z}) \leq p_{Z_1}(\boldsymbol{z})$.*

This intuitively makes sense – given some representation $\boldsymbol{z}$, the adversary computes likelihood under both distributions $\mathcal{Z}_0$ and $\mathcal{Z}_1$, and predicts the attribute with higher likelihood for that $\boldsymbol{z}$. Liao et al.

(2019) also observed that the optimal adversary can be phrased as $\arg\max_a p(a|\boldsymbol{z})$. So far, prior work mostly focused on mapping input $\boldsymbol{x}$ to the latent representation $\boldsymbol{z} = f_\theta(\boldsymbol{x}, a)$ via standard neural networks. However, for such models, given densities $p_0(\boldsymbol{x})$ and $p_1(\boldsymbol{x})$ over the input space, it is intractable to compute the densities $p_{Z_0}(\boldsymbol{z})$ and $p_{Z_1}(\boldsymbol{z})$ in the latent space as many inputs $\boldsymbol{x}$ can be mapped to the same latent $\boldsymbol{z}$ and we cannot use inverse function theorem. Consequently, adversarial training methods cannot compute the optimal adversary and thus resort to a lower bound.

**Encoding with normalizing flows**    Our approach, named Fair Normalizing Flows (FNF), consists of two models, $f_0$ and $f_1$, that encode inputs from the groups with sensitive attributes $a = 0$ and $a = 1$, respectively. We show a high-level overview of FNF in Fig. 1. Note that models $f_0$ and $f_1$ are parameterized by $\theta_0$ and $\theta_1$, but we do not write this explicitly to ease the notation. Given some input $\boldsymbol{x}_0 \sim p_0$, it is encoded to $\boldsymbol{z}_0 = f_0(\boldsymbol{x}_0)$, inducing a probability distribution $\mathcal{Z}_0$ with density $p_{Z_0}(\boldsymbol{z})$ over all possible latent representations $\boldsymbol{z}$. Similarly, inputs $\boldsymbol{x}_1 \sim p_1$ are encoded to $\boldsymbol{z}_1 = f_1(\boldsymbol{x}_1)$, inducing the probability distribution $\mathcal{Z}_1$ with density $p_{Z_1}(\boldsymbol{z})$. Clearly, if we can train $f_0$ and $f_1$ so that the resulting distributions $\mathcal{Z}_0$ and $\mathcal{Z}_1$ have small distance, then we can guarantee fairness of the representations using the bounds from Madras et al. (2018). As evaluating the statistical distance is intractable for most neural networks, we need a model family that allows us to compute this quantity.

We propose to use bijective encoders $f_0$ and $f_1$ based on normalizing flows (Rezende & Mohamed, 2015) which allow us to compute the densities at $\boldsymbol{z}$ using the change of variables formula

$$\log p_{Z_a}(\boldsymbol{z}) = \log p_a(f_a^{-1}(\boldsymbol{z})) + \log \left| \det \frac{\partial f_a^{-1}(\boldsymbol{z})}{\partial \boldsymbol{z}} \right| \tag{4}$$

for $a \in \{0, 1\}$. Recall that Lemma 5.1 provides a form of the optimal adversary. To compute the statistical distance it remains to evaluate the expectations $\mathbb{E}_{\boldsymbol{z} \sim \mathcal{Z}_0}[\mu^*(\boldsymbol{z})]$ and $\mathbb{E}_{\boldsymbol{z} \sim \mathcal{Z}_1}[\mu^*(\boldsymbol{z})]$. Sampling from $\mathcal{Z}_0$ and $\mathcal{Z}_1$ is straightforward – we can sample $\boldsymbol{x}_0 \sim p_0$ and $\boldsymbol{x}_1 \sim p_1$, and then pass the samples $\boldsymbol{x}_0$ and $\boldsymbol{x}_1$ through the respective encoders $f_0$ and $f_1$ to obtain $\boldsymbol{z}_0 \sim \mathcal{Z}_0$ and $\boldsymbol{z}_1 \sim \mathcal{Z}_1$. Given that the outputs of $\mu^*$ are bounded between 0 and 1, we can then use Hoeffding inequality to compute the confidence intervals for our estimate using a finite number of samples.

**Lemma 5.2.** *Given a finite number of samples $\boldsymbol{x}_0^1, \boldsymbol{x}_0^2, ..., \boldsymbol{x}_0^n \sim p_0$ and $\boldsymbol{x}_1^1, \boldsymbol{x}_1^2, ..., \boldsymbol{x}_1^n \sim p_1$, denote as $\boldsymbol{z}_0^i = f_0(\boldsymbol{x}_0^i)$ and $\boldsymbol{z}_1^i = f_1(\boldsymbol{x}_1^i)$ and let $\hat{\Delta}(p_{Z_0}, p_{Z_1}) := |\frac{1}{n}\sum_{i=1}^n \mu^*(\boldsymbol{z}_0^i) - \frac{1}{n}\sum_{i=1}^n \mu^*(\boldsymbol{z}_1^i)|$ be an empirical estimate of the statistical distance $\Delta(p_{Z_0}, p_{Z_1})$. Then, for $n \geq -2\log\left(\frac{1-\sqrt{1-\delta}}{2}\right)/\epsilon^2$ we are guaranteed that $\Delta(p_{Z_0}, p_{Z_1}) \leq \hat{\Delta}(p_{Z_0}, p_{Z_1}) + \epsilon$ with probability at least $1 - \delta$.*

**Training flow-based encoders**    The next challenge is to design a training procedure for our newly proposed architecture. The main issue is that the statistical distance is not differentiable (as the classifier $\mu^*$ is binary), so we replace it with a differentiable proxy based on the symmetrized KL divergence, shown in Lemma 5.3 below (proof provided in Appendix A.1). We show a high-level description of our training procedure in Algorithm 1. In each step, we sample a batch of $\boldsymbol{x}_0$ and $\boldsymbol{x}_1$ from the respective distributions and encode them to the representations $\boldsymbol{z}_0$ and $\boldsymbol{z}_1$. We then estimate the symmetrized KL divergence between distributions $\mathcal{Z}_0$ and $\mathcal{Z}_1$, denoted as $\mathcal{L}_0 + \mathcal{L}_1$, and combine it with a classification loss $\mathcal{L}_{clf}$ using tradeoff parameter $\gamma$, and perform a gradient descent step to minimize the joint loss.

---

**Algorithm 1** Learning Fair Normalizing Flows

**Input:** $N, B, \gamma, p_0, p_1$
Initialize $h, f_0, f_1$ with parameters $\theta_h, \theta_0, \theta_1$
**for** $i = 1$ **to** $N$ **do**
    **for** $j = 1$ **to** $B$ **do**
        Sample $\boldsymbol{x}_0^j \sim p_0, \boldsymbol{x}_1^j \sim p_1$
        $\boldsymbol{z}_0^j = f_0(\boldsymbol{x}_0^j)$
        $\boldsymbol{z}_1^j = f_1(\boldsymbol{x}_1^j)$
    **end for**
    $\mathcal{L}_0 = \frac{1}{B}\sum_{j=1}^B (\log p_{Z_0}(\boldsymbol{z}_0^j) - \log p_{Z_1}(\boldsymbol{z}_0^j))$
    $\mathcal{L}_1 = \frac{1}{B}\sum_{j=1}^B (\log p_{Z_1}(\boldsymbol{z}_1^j) - \log p_{Z_0}(\boldsymbol{z}_1^j))$
    $\mathcal{L} = \gamma(\mathcal{L}_0 + \mathcal{L}_1) + (1 - \gamma)\mathcal{L}_{clf}$
    Update $\theta_a \leftarrow \theta_a - \alpha\nabla_{\theta_a}\mathcal{L}$, for $a \in \{0, 1\}$
    Update $\theta_h \leftarrow \theta_h - \alpha\nabla_{\theta_h}\mathcal{L}$
**end for**

---

While we use a convex scalarization scheme to obtain the joint loss in Algorithm 1, our approach is independent of the concrete multi-objective optimization objective (see Appendix C).

**Lemma 5.3.** *We can bound $\Delta(p_{Z_0}, p_{Z_1})^2 \leq \frac{1}{4}(KL(p_{Z_0}, p_{Z_1}) + KL(p_{Z_1}, p_{Z_0}))$.*

**Bijective encoders for categorical data**    Many fairness datasets consist of categorical data, and often even continuous data is discretized before training. In this case, we will show that the optimal

bijective representation can be easily computed. Consider the case of discrete samples $\boldsymbol{x}$ coming from a probability distribution $p(\boldsymbol{x})$ where each component $x_i$ takes a value from a finite set $\{1, 2, \ldots, d_i\}$. Similar to the continuous case, our goal is to find bijections $f_0$ and $f_1$ that minimize the statistical distance of the latent distributions. Intuitively, we want to pair together inputs that have similar probabilities in both $p_0$ and $p_1$. In Lemma 5.4 we show that the solution that minimizes the statistical distance is obtained by sorting the inputs according to their probabilities in $p_0$ and $p_1$, and then matching inputs at the corresponding indices in these two sorted arrays. As this can result in a bad classification accuracy when inputs with different target labels get matched together, we can obtain another representation by splitting inputs in two groups according to the predicted classification label and then matching inputs in each group using Lemma 5.4. We can trade off accuracy and fairness by randomly selecting one of the two mappings based on a parameter $\gamma$.

**Lemma 5.4.** *Let $\mathcal{X} = \{\boldsymbol{x}_1, ..., \boldsymbol{x}_m\}$ and bijections $f_0, f_1 : \mathcal{X} \to \mathcal{X}$. Denote $i_1, i_2, ..., i_m$ and $j_1, ..., j_m$ permutations of $\{1, 2, ..., m\}$ such that $p_0(\boldsymbol{x}_{i_1}) \leq p_0(\boldsymbol{x}_{i_2}) \leq ... \leq p_0(\boldsymbol{x}_{i_m})$ and $p_1(\boldsymbol{x}_{j_1}) \leq p_1(\boldsymbol{x}_{j_2}) \leq ... \leq p_1(\boldsymbol{x}_{j_m})$. The encoders defined by mapping $f_0(\boldsymbol{x}_k) = \boldsymbol{x}_k$ and $f_1(\boldsymbol{x}_{j_k}) = \boldsymbol{x}_{i_k}$ are bijective representations with the smallest possible statistical distance.*

**Statistical distance of true vs. estimated density** In this work we assume access to a density of the inputs for both groups and we *provably guarantee* fairness with respect to *this* density. While it is sensible in the cases where the density estimate can be trusted (e.g., if it was provided by a regulatory agency), in many practical scenarios, and our experiments in Section 6, we only have an estimate $\hat{p}_0$ and $\hat{p}_1$ of the true densities $p_0$ and $p_1$. We now want to know how far off our guarantees are compared to the ones for the true density. The following theorem provides a way to theoretically bound the statistical distance between $p_{Z_0}$ and $p_{Z_1}$ using the statistical distance between $\hat{p}_{Z_0}$ and $\hat{p}_{Z_1}$.

**Theorem 5.5.** *Let $\hat{p}_0$ and $\hat{p}_1$ be density estimates such that $TV(\hat{p}_0, p_0) < \epsilon/2$ and $TV(\hat{p}_1, p_1) < \epsilon/2$, where $TV$ stands for the total variation between two distributions. If we denote the latent distributions $f_0(\hat{p}_0)$ and $f_1(\hat{p}_1)$ as $\hat{p}_{Z_0}$ and $\hat{p}_{Z_1}$ then $\Delta(p_{Z_0}, p_{Z_1}) \leq \Delta(\hat{p}_{Z_0}, \hat{p}_{Z_1}) + \epsilon$.*

This theorem can be combined with Lemma 5.2 to obtain a high probability upper bound on the statistical distance of the underlying true densities using estimated densities and a finite number of samples. Computing exact constants for the theorem is often not tractable, but as we will show experimentally, in practice the bounds computed on the estimated distribution in fact bound adversarial accuracy on the true distribution. Moreover, for low-dimensional data relevant to fairness, obtaining good estimates can be provably done for models such as Gaussian Mixture Models (Hardt & Price, 2015) and Kernel Density Estimation (Jiang, 2017). We can thus leverage the rich literature on density estimation (Rezende & Mohamed, 2015; Dinh et al., 2016; van den Oord et al., 2016a;b;c) to estimate $\hat{p}_0$ and $\hat{p}_1$. Importantly, FNF is agnostic to the density estimation method (as we show in Appendix C), and can benefit from future advances in the field. Finally, we note that density estimation has already been applied in a variety of security-critical areas such as fairness (Song et al., 2019), adversarial robustness (Wong & Kolter, 2020), and anomaly detection (Pidhorskyi et al., 2018).

## 6 EXPERIMENTAL EVALUATION

In this section, we evaluate Fair Normalizing Flows (FNF) on several standard datasets from the fairness literature. We consider UCI Adult and Crime (Dua & Graff, 2017), Compas (Angwin et al., 2016), Law School (Wightman, 2017), and the Health Heritage dataset. We preprocess Compas and Adult into categorical datasets by discretizing continuous features, and we keep the other datasets as continuous. Moreover, we preprocess the datasets by dropping uninformative features, facilitating the learning of a good density estimate, while keeping accuracy high (details shown in Appendix B). We make all of our code publicly available at `https://github.com/eth-sri/fnf`.

**Evaluating Fair Normalizing Flows** We first evaluate FNF's effectiveness in learning fair representations by training different FNF models with different values for the utility vs. fairness tradeoff parameter $\gamma$. We estimate input densities using RealNVP (Dinh et al., 2016) for Health, MADE (Germain et al., 2015) for Adult and Compas, and Gaussian Mixture Models (GMMs) for the rest (we experiment with other density estimation methods in Appendix C). For continuous datasets we use RealNVP as encoder, while for categorical datasets we compute the optimal bijective representations using Lemma 5.4. Fig. 4 shows our results, each point representing a single model, with models on

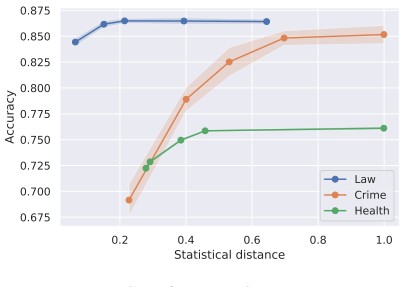

(a) Continuous datasets

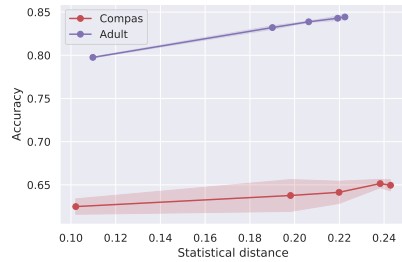

(b) Categorical datasets

Figure 4: Fair Normalizing Flows (FNF) on continuous and categorical data. The points show different accuracy vs. statistical distance tradeoffs (with 95% confidence intervals from varied random seeds), demonstrating that FNF significantly reduces statistical distance while retaining high accuracy.

Table 1: Adversarial fair representation learning methods are only fair w.r.t. adversaries from a training family $\mathcal{G}$ while FNF provides a provable upper bound on the maximum accuracy of *any* adversary.

|  |  | Adv Acc | | |
| --- | --- | --- | --- | --- |
|  | Acc | $g \in \mathcal{G}$ | $g \notin \mathcal{G}$ | Max Adv Acc |
| ADV FORGETTING (Jaiswal et al., 2020) | 85.99 | 66.68 | 74.50 | ✗ |
| MAXENT-ARL (Roy & Boddeti, 2019) | 85.90 | 50.00 | 85.18 | ✗ |
| LAFTR (Madras et al., 2018) | **86.09** | 72.05 | 84.58 | ✗ |
| FNF (our work) | 84.43 | N/A | **59.56** | **61.12** |

the right focusing on classification accuracy, and models on the left gradually increasing their fairness focus. The results in Fig. 4, averaged over 5 random seeds, indicate that FNF successfully reduces the statistical distance between representations of sensitive groups while maintaining high accuracy. We observe that for some datasets (e.g., Law School) enforcing fairness only slightly degrades accuracy, while for others there is a substantial drop (e.g., Crime). In such datasets where the label and sensitive attribute are highly correlated we cannot achieve fairness and high accuracy simultaneously (Menon & Williamson, 2018; Zhao & Gordon, 2019). Overall, we see that FNF is generally insensitive to the random seed and can reliably enforce fairness. Recall that we have focused on minimizing statistical distance of learned representations because, as mentioned earlier, Madras et al. (2018) have shown that fairness metrics such as demographic parity, equalized odds and equal opportunity can all be bounded by statistical distance. For example, FNF reduces the demographic parity distance of a classifier on Health from 0.39 to 0.08 with an accuracy drop of 3.9% (we provide similar results showing FNF's good performance for equalized odds and equality of opportunity in Appendix C).

**Bounding adversarial accuracy** Recall that the guarantees provided by FNF hold for estimated densities $\hat{p}_0$ and $\hat{p}_1$. Namely, the maximum adversarial accuracy for predicting whether the latent representation $z$ originates from distribution $\hat{\mathcal{Z}}_0$ or $\hat{\mathcal{Z}}_1$ is bounded by $(1+\Delta(\hat{p}_{Z_0}, \hat{p}_{Z_1}))/2$. In this experiment, we investigate how well these guarantees transfer to the underlying distributions $\mathcal{Z}_0$ and $\mathcal{Z}_1$. In Fig. 5 we show our upper bound on the adversarial accuracy computed from the statistical distance using the estimated densities (diagonal dashed line), together with adversarial accuracies obtained by training an adversary, a multilayer perceptron (MLP) with two hidden layers of 50 neurons, for

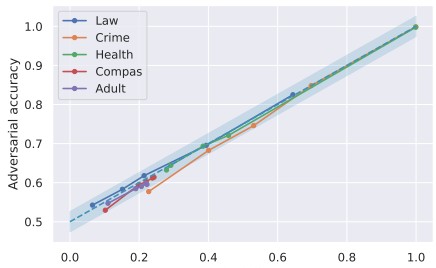

Figure 5: Bounding adversarial accuracy.

each model from Fig. 4. We also show 95% confidence intervals obtained using the Hoeffding bound from Lemma 5.2. We observe that our upper bound from the estimated densities $\hat{p}_0$ and $\hat{p}_1$ provides a tight upper bound on the adversarial accuracy for the true distributions $\mathcal{Z}_0$ and $\mathcal{Z}_1$. This demonstrates that, even though the exact constants from Theorem 5.5 are intractable, our density estimate is good enough in practice, and our bounds hold for adversaries on the true distribution.

**Comparison with adversarial training**  We now compare FNF with adversarial fair representation learning methods on Adult dataset: LAFTR-DP ($\gamma = 2$) (Madras et al., 2018), MaxEnt-ARL ($\alpha = 10$) (Roy & Boddeti, 2019), and Adversarial Forgetting ($\rho = 0.001, \delta = 1, \lambda = 0.1$) (Jaiswal et al., 2020). We train with a family of adversaries $\mathcal{G}$ trying to predict the sensitive attribute from the latent representation. Here, the families $\mathcal{G}$ are MLPs with 1 hidden layer of 8 neurons for LAFTR-DP, and 2 hidden layers with 64 neurons and 50 neurons for MaxEnt-ARL and Adversarial Forgetting, respectively. In Table 1 we show that these methods generally prevent adversaries from $\mathcal{G}$ to predict the sensitive attributes. However, we can still attack these representations using either larger MLPs (3 layers of 200 neurons for LAFTR-DP) or simple preprocessing steps (for MaxEnt-ARL and Adversarial Forgetting) as proposed by Gupta et al. (2021) (essentially reproducing their results). Our results confirm findings from prior work (Feng et al., 2019; Xu et al., 2020; Gupta et al., 2021): adversarial training provides no guarantees against adversaries outside $\mathcal{G}$. In contrast, FNF computes a provable upper bound on the accuracy of *any* adversary for the estimated input distribution, and Table 1 shows that this extends to the true distribution. FNF thus learns representations with significantly lower adversarial accuracy with only minor decrease in task accuracy.

**Algorithmic recourse with FNF**  We next experiment with FNF's bijectivity to perform recourse, i.e., reverse an unfavorable outcome, which is considered to be fundamental to explainable algorithmic decision-making (Venkatasubramanian & Alfano, 2020). To that end, we apply FNF with $\gamma = 1$ to the Law School dataset with three features: LSAT score, GPA, and the college to which the student applied (ordered decreasingly in admission rate). For all rejected applicants, i.e., $\boldsymbol{x}$ such that $h(f_a(\boldsymbol{x})) = h(\boldsymbol{z}) = 0$, we compute the closest $\tilde{\boldsymbol{z}}$ (corresponding to a point $\tilde{\boldsymbol{x}}$ from the dataset) w.r.t. the $\ell_2$-distance in latent space such that $h(\tilde{\boldsymbol{z}}) = 1$. We then linearly interpolate between $\boldsymbol{z}$ and $\tilde{\boldsymbol{z}}$ to find a (potentially crude) approximation of the closest point to $\boldsymbol{z}$ in latent space with positive prediction. Using the bijectivity of our encoders, we can compute the corresponding average feature change in the original space that would have caused a positive decision: increasing LSAT by $4.2$ (non-whites) and $7.7$ (whites), and increasing GPA by $0.7$ (non-whites) and $0.6$ (whites), where we only report recourse in the cases where the college does not change since this may not be actionable advice for certain applicants (Zhang et al., 2018; Ustun et al., 2019; Poyiadzi et al., 2020). In Appendix C, we also show that, unlike prior work, FNF enables practical interpretability analyses.

**Flow architectures**  In the next experiment we compare the RealNVP encoder with an alternative encoder based on the Neural Spline Flows architecture (Durkan et al., 2019) for the Crime dataset. In Table 2 we show the statistical distance and accuracy for models obtained using different values for the tradeoff parameter $\gamma$. We can observe that both flows offer similar performance. Note that FNF will benefit from future advances in normalizing flows research, as it is orthogonal to the concrete flow architecture that is used for training.

Table 2: FNF performance with different flow encoder architectures.

| | RealNVP | | NSF | |
|---|---|---|---|---|
| $\gamma$ | $\Delta$ | Acc | $\Delta$ | Acc |
| 0.00 | 1.00 | 0.85 | 1.00 | 0.84 |
| 0.02 | 0.70 | 0.85 | 0.71 | 0.85 |
| 0.10 | 0.53 | 0.83 | 0.54 | 0.83 |
| 0.90 | 0.23 | 0.69 | 0.24 | 0.69 |

**Transfer learning**  Unlike prior work, transfer learning with FNF requires no additional reconstruction loss since both encoders are invertible and thus preserve all information about the input data. To demonstrate this, we follow the setup from Madras et al. (2018) and train a model to predict the Charlson Index for the Health Heritage Prize dataset. We then transfer the learned encoder and train a classifier for the task of predicting the primary condition group. Our encoder reduces the statistical distance from 0.99 to 0.31 (this is independent of the label). For the primary condition group MSC2a3 we retain the accuracy at 73.8%, while for METAB3 it slightly decreases from 75.4% to 73.1%.

## 7 CONCLUSION

We introduced Fair Normalizing Flows (FNF), a new method for learning representations ensuring that no adversary can predict sensitive attributes at the cost of a small accuracy decrease. This guarantee is stronger than prior work which only considers adversaries from a restricted model family. The key idea is to use an encoder based on normalizing flows which allows computing the exact likelihood in the latent space, given an estimate of the input density. Our experimental evaluation on several datasets showed that FNF effectively enforces fairness without significantly sacrificing utility, while simultaneously allowing interpretation of the representations and transferring to unseen tasks.

ETHICS STATEMENT

Since machine learning models have been shown to reinforce the human biases that are embedded in the training data, regulators and scientists alike are striving to propose novel regulations and algorithms to ensure the fairness of such models. Our method enables data producers to learn fair data representations that are guaranteed to be non-discriminatory regardless of the concrete downstream use case. Since our method relies on accurate density estimates, we envision that the data regulators, whose tasks already include determining fairness criteria, data sources, and auditing results, would create a regulatory framework for density estimation that can then be realized by, e.g., industrial partners. Importantly, this would not require regulators to estimate the densities themselves. Nevertheless, data regulators would need to take great care when formulating such legislation since the potential negative effects of poor density estimates are still largely unexplored, both in the context of our work and in the broader field (particularly for high-dimensional data). In this setting, any data producer using our method would then be able to guarantee the fairness of all potential downstream consumer models.

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

## A   APPENDIX

### A.1   PROOFS

Here we present the proofs of the theorems used in the paper.

**Proof of Lemma 5.1 (Optimal adversary)**

*Proof.* Here we prove Lemma 5.1, which states the form of an adversary $\mu \colon \mathbb{R}^d \to \{0, 1\}$ that can achieve the highest discrimination between $\mathcal{Z}_0$ and $\mathcal{Z}_1$. We start by rewriting the equation for statistical distance $\Delta(p_{Z_0}, p_{Z_1})$ as follows:

$$\Delta(p_{Z_0}, p_{Z_1}) = \sup_{\mu} \left| \mathbb{E}_{\boldsymbol{z} \sim \mathcal{Z}_0}[\mu(\boldsymbol{z})] - \mathbb{E}_{\boldsymbol{z} \sim \mathcal{Z}_1}[\mu(\boldsymbol{z})] \right| = \sup_{\mu} \left| \int_{\boldsymbol{z}} (p_{Z_0}(\boldsymbol{z}) - p_{Z_1}(\boldsymbol{z}))\mu(\boldsymbol{z}) \, dz \right|. \quad (5)$$

To maximize the absolute value of the integral, we either have to maximize or minimize the integral $\int_{\boldsymbol{z}} (p_{Z_0}(\boldsymbol{z}) - p_{Z_1}(\boldsymbol{z}))\mu(\boldsymbol{z}) \, dz$. Clearly, to maximize the integral we should choose the function $\mu = \mathbb{1}_{\{p_{Z_0}(\boldsymbol{z}) \geq p_{Z_1}(\boldsymbol{z})\}}$, for which $\mu(\boldsymbol{z}) = 1$ if and only if $p_{Z_0}(\boldsymbol{z}) \geq p_{Z_1}(\boldsymbol{z})$, and 0 otherwise. Similarly, to minimize the integral we should choose the function $\mu = \mathbb{1}_{\{p_{Z_0}(\boldsymbol{z}) \leq p_{Z_1}(\boldsymbol{z})\}}$ for which $\mu(\boldsymbol{z}) = 1$ if and only if $p_{Z_0}(\boldsymbol{z}) \leq p_{Z_1}(\boldsymbol{z})$, and 0 otherwise. In fact, it can be easily observed that both options result in the same absolute value of the integral so we can, without loss of generality, choose the option $\mu^*(\boldsymbol{z}) = \mathbb{1}_{\{p_{Z_0}(\boldsymbol{z}) \leq p_{Z_1}(\boldsymbol{z})\}}$. This subsequently yields $\Delta(p_{Z_0}, p_{Z_1}) = \left| \mathbb{E}_{\boldsymbol{z} \sim \mathcal{Z}_0}[\mu^*(\boldsymbol{z})] - \mathbb{E}_{\boldsymbol{z} \sim \mathcal{Z}_1}[\mu^*(\boldsymbol{z})] \right|$. Moreover, we can also write $\Delta(p_{Z_0}, p_{Z_1}) = \left| \int_{\boldsymbol{z}} (p_{Z_1}(\boldsymbol{z}) - p_{Z_0}(\boldsymbol{z}))\mu^*(\boldsymbol{z}) \, dz \right| = \left| \int_{\boldsymbol{z}} \max(0, p_{Z_1}(\boldsymbol{z}) - p_{Z_0}(\boldsymbol{z})) \, dz \right|$ (this will be used to prove Theorem 5.5). $\square$

**Proof of Lemma 5.2 (Finite sample estimate)**

*Proof.* We start by plugging in the optimal adversary $\mu^*$ from Lemma 5.1 into the definition of the statistical distance. Let $\mu^*$ be the optimal adversary defined in Lemma 5.1, namely $\mu^*(\boldsymbol{z}) = 1$ if and only if $p_{Z_0}(\boldsymbol{z}) < p_{Z_1}(\boldsymbol{z})$. We first write the statistical distance in terms of the optimal adversary $\mu^*$, then bound the statistical distance using triangle inequality and finally apply Hoeffding's inequality on the individual terms:

$$\begin{aligned}
\Delta(\mathcal{Z}_0, \mathcal{Z}_1) &= \sup_{\mu} \left| \mathbb{E}_{\boldsymbol{z} \sim \mathcal{Z}_0}[\mu(\boldsymbol{z})] - \mathbb{E}_{\boldsymbol{z} \sim \mathcal{Z}_1}[\mu(\boldsymbol{z})] \right| \\
&= \left| \mathbb{E}_{\boldsymbol{z} \sim \mathcal{Z}_0}[\mu^*(\boldsymbol{z})] - \mathbb{E}_{\boldsymbol{z} \sim \mathcal{Z}_1}[\mu^*(\boldsymbol{z})] \right| \\
&= \left| \mathbb{E}_{\boldsymbol{z} \sim \mathcal{Z}_0}[\mu^*(\boldsymbol{z})] - \frac{1}{n}\sum_{i=1}^n \mu^*(\boldsymbol{z}_0^i) + \frac{1}{n}\sum_{i=1}^n \mu^*(\boldsymbol{z}_0^i) - \frac{1}{n}\sum_{i=1}^n \mu^*(\boldsymbol{z}_1^i) + \frac{1}{n}\sum_{i=1}^n \mu^*(\boldsymbol{z}_1^i) - \mathbb{E}_{\boldsymbol{z} \sim \mathcal{Z}_1}[\mu^*(\boldsymbol{z})] \right| \\
&\leq \left| \mathbb{E}_{\boldsymbol{z} \sim \mathcal{Z}_0}[\mu^*(\boldsymbol{z})] - \frac{1}{n}\sum_{i=1}^n \mu^*(\boldsymbol{z}_0^i) \right| + \hat{\Delta}(\mathcal{Z}_0, \mathcal{Z}_1) + \left| \frac{1}{n}\sum_{i=1}^n \mu^*(\boldsymbol{z}_1^i) - \mathbb{E}_{\boldsymbol{z} \sim \mathcal{Z}_1}[\mu^*(\boldsymbol{z})] \right| \\
&\leq \hat{\Delta}(\mathcal{Z}_0, \mathcal{Z}_1) + \epsilon
\end{aligned}$$

where Hoeffding's inequality guarantees that with probability at least $(1 - 2\exp(-n\epsilon^2/2))^2 \geq 1 - \delta$ the first and last summand are at most $\epsilon/2$. $\square$

**Proof of Lemma 5.3 (Bounding the statistical distance with symmetrized KL)**

*Proof.* We can bound the statistical distance $\Delta(p_{Z_0}, p_{Z_1})$ by noticing that $|\mu(\boldsymbol{z})| \leq 1$ because $\mu$ is a binary classifier and thus

$$\Delta(p_{Z_0}, p_{Z_1}) = \sup_{\mu} \left| \int_{\boldsymbol{z}} (p_{Z_0}(\boldsymbol{z}) - p_{Z_1}(\boldsymbol{z}))\mu(\boldsymbol{z}) \, dz \right| \leq \int_{\boldsymbol{z}} |p_{Z_0}(\boldsymbol{z}) - p_{Z_1}(\boldsymbol{z})| \, dz = TV(p_{Z_0}, p_{Z_1}).$$

Pinsker's inequality guarantees that $TV(p_{Z_0}, p_{Z_1})^2 \leq \frac{1}{2}KL(p_{Z_0}, p_{Z_1})$. Given that $TV(p_{Z_0}, p_{Z_1}) = TV(p_{Z_1}, p_{Z_0})$ we also have that $TV(p_{Z_0}, p_{Z_1})^2 \leq \frac{1}{2}KL(p_{Z_1}, p_{Z_0})$. Thus, in order to bound the statistical distance we can use $2\Delta(p_{Z_0}, p_{Z_1})^2 \leq 2TV(p_{Z_0}, p_{Z_1})^2 \leq \frac{1}{2}KL(p_{Z_0}, p_{Z_1}) + \frac{1}{2}KL(p_{Z_1}, p_{Z_0})$, which corresponds to our objective with symmetrized KL divergence used in Algorithm 1. $\square$

**Proof of Lemma 5.4 (Encoding for discrete distributions)**

*Proof.* Without loss of generality we can assume that $f_0(\boldsymbol{x}_k) = \boldsymbol{x}_k$. Assume that $f_1(\boldsymbol{x}) = \boldsymbol{y}$ and $f_1(\boldsymbol{x}') = \boldsymbol{y}'$ where $p_0(\boldsymbol{x}) < p_0(\boldsymbol{x}')$ and $p_1(\boldsymbol{y}) > p_1(\boldsymbol{y}')$. Let $d_1 = |p_0(\boldsymbol{x}) - p_1(\boldsymbol{y})| + |p_0(\boldsymbol{x}') - p_1(\boldsymbol{y}')|$ and $d_2 = |p_0(\boldsymbol{x}) - p_1(\boldsymbol{y}')| + |p_0(\boldsymbol{x}') - p_1(\boldsymbol{y})|$. In the case when $p_1(\boldsymbol{y}) \leq p_0(\boldsymbol{x})$ or $p_1(\boldsymbol{y}') \geq p_0(\boldsymbol{x}')$ it is easy to show that $d_1 = d_2$. Now, in the case when $p_0(\boldsymbol{x}) < p_1(\boldsymbol{y}') < p_0(\boldsymbol{x}') < p_1(\boldsymbol{y})$, we can see that $d_2 \leq d_1 - (p_0(\boldsymbol{x}) - p_1(\boldsymbol{y}')) < d_1$. Similarly, when $p_0(\boldsymbol{x}) < p_1(\boldsymbol{y}') < p_1(\boldsymbol{y}) < p_0(\boldsymbol{x}')$, we can see that $d_2 \leq d_1 - (p_1(\boldsymbol{y}) - p_1(\boldsymbol{y}')) < d_1$. In all cases, $d_2 \leq d_1$, which means that if we swap $f_1(\boldsymbol{x})$ and $f_1(\boldsymbol{x}')$ the total variation either decreases or stays the same. We can repeatedly swap such pairs, and once there are no more swaps to do, we arrive at the condition of the lemma where the two arrays are sorted the same way. $\qquad\square$

**Proof of Theorem 5.5 (True and approximate distributions)**

*Proof.* Let $a \in \{0, 1\}$ be a sensitive attribute, and assume that $TV(\hat{p}_a, p_a) < \epsilon/2$. Recall that the change of variables for probability densities gives us that for the representation $\boldsymbol{z} = f_a(\boldsymbol{x})$ we have, similar as in Eq. (4), $p_{Z_a}(\boldsymbol{z}) = p_a(\boldsymbol{x}) \left| \det \frac{\partial f_a(\boldsymbol{x})}{\partial \boldsymbol{x}} \right|^{-1}$ and $\hat{p}_{Z_a}(\boldsymbol{z}) = \hat{p}_a(\boldsymbol{x}) \left| \det \frac{\partial f_a(\boldsymbol{x})}{\partial \boldsymbol{x}} \right|^{-1}$. Then, using those formulas together with the change of variables rule for multivariate integrals we derive:

$$
\begin{aligned}
TV(\hat{p}_{Z_a}, p_{Z_a}) &= \int_{\boldsymbol{z}} |\hat{p}_{Z_a}(\boldsymbol{z}) - p_{Z_a}(\boldsymbol{z})| \; dz \\
&= \int_{\boldsymbol{x}} |\hat{p}_{Z_a}(f_a(\boldsymbol{x})) - p_{Z_a}(f_a(\boldsymbol{x}))| \left| \det \frac{\partial f_a(\boldsymbol{x})}{\partial \boldsymbol{x}} \right| \; dx \\
&= \int_{\boldsymbol{x}} \left| \hat{p}_a(\boldsymbol{x}) \left| \det \frac{\partial f_a(\boldsymbol{x})}{\partial \boldsymbol{x}} \right|^{-1} - p_a(\boldsymbol{x}) \left| \det \frac{\partial f_a(\boldsymbol{x})}{\partial \boldsymbol{x}} \right|^{-1} \right| \left| \det \frac{\partial f_a(\boldsymbol{x})}{\partial \boldsymbol{x}} \right| \; dx \\
&= \int_{\boldsymbol{x}} |\hat{p}_a(\boldsymbol{x}) - p_a(\boldsymbol{x})| \; dx \\
&= TV(\hat{p}_a, p_a) \\
&< \epsilon/2.
\end{aligned}
$$

We will also use the observation from the proof of Lemma 5.1 which states that

$$
\Delta(p_{Z_0}, p_{Z_1}) = \left| \int_{\boldsymbol{z}} \max(0, p_{Z_1}(\boldsymbol{z}) - p_{Z_0}(\boldsymbol{z})) \, dz \right|.
$$

We can now observe that the following inequality holds:

$$
\max(0, p_{Z_0}(\boldsymbol{z}) - p_{Z_1}(\boldsymbol{z})) \leq |p_{Z_0}(\boldsymbol{z}) - \hat{p}_{Z_0}(\boldsymbol{z})| + \max(0, \hat{p}_{Z_0}(\boldsymbol{z}) - \hat{p}_{Z_1}(\boldsymbol{z})) + |\hat{p}_{Z_1}(\boldsymbol{z}) - p_{Z_1}(\boldsymbol{z})|
$$

Integrating both sides yields:

$$
\begin{aligned}
\Delta(p_{Z_0}, p_{Z_1}) &\leq TV(p_{Z_0}(\boldsymbol{z}), \hat{p}_{Z_0}(\boldsymbol{z})) + \Delta(\hat{p}_{Z_0}, \hat{p}_{Z_1}) + TV(p_{Z_1}(\boldsymbol{z}), \hat{p}_{Z_1}(\boldsymbol{z})) \\
&\leq \Delta(\hat{p}_{Z_0}, \hat{p}_{Z_1}) + \epsilon
\end{aligned}
$$

Here, the last inequality follows from the previously proved inequality $TV(\hat{p}_{Z_a}, p_{Z_a}) < \epsilon/2$, applied for both $a = 0$ and $a = 1$. $\qquad\square$

## B  EXPERIMENTAL SETUP

In this section, we provide the full specification of our experimental setup. We first discuss the datasets considered and the corresponding preprocessing methods employed. We empirically validate that our preprocessing maintains high accuracy on the respective prediction tasks. Finally, we specify the hyperparameters and computing resources for the experiments in Section 6.

Table 3: Statistics for train, validation, and test datasets. In general, the label ($y$) and sensitive attribute ($a$) distributions are highly skewed, which is why we report balanced accuracy.

|        |            | SIZE    | $a = 1$ | $y = 1 \mid a = 0$ | $y = 1 \mid a = 1$ | $y = 1$ |
|--------|------------|---------|---------|--------------------|--------------------|---------|
| ADULT  | TRAIN      | 24 129  | 32.6%   | 28.5%              | 21.6%              | 24.9%   |
|        | VALIDATION | 6033    | 31.9%   | 28.3%              | 19.9%              | 24.9%   |
|        | TEST       | 15 060  | 32.6%   | 27.9%              | 19.1%              | 24.6%   |
| COMPAS | TRAIN      | 3377    | 60.6%   | 60.9%              | 46.8%              | 52.3%   |
|        | VALIDATION | 845     | 60.0%   | 60.1%              | 46.9%              | 52.2%   |
|        | TEST       | 1056    | 58.9%   | 61.8%              | 51.3%              | 55.6%   |
| CRIME  | TRAIN      | 1276    | 42.3%   | 71.3%              | 17.8%              | 48.7%   |
|        | VALIDATION | 319     | 40.8%   | 78.8%              | 21.5%              | 55.5%   |
|        | TEST       | 399     | 43.1%   | 74.9%              | 16.3%              | 49.6%   |
| HEALTH | TRAIN      | 139 785 | 35.7%   | 79.0%              | 48.3%              | 68.0%   |
|        | VALIDATION | 34 947  | 35.6%   | 79.3%              | 49.2%              | 68.6%   |
|        | TEST       | 43 683  | 35.4%   | 78.8%              | 48.3%              | 68.0%   |
| LAW    | TRAIN      | 55 053  | 18.0%   | 28.5%              | 21.6%              | 27.3%   |
|        | VALIDATION | 13 764  | 17.5%   | 28.3%              | 19.9%              | 26.8%   |
|        | TEST       | 17 205  | 17.5%   | 27.9%              | 19.1%              | 26.3%   |

## B.1 DATASETS

We consider five commonly studied datasets from the fairness literature: Adult and Crime from the UCI machine learning repository (Dua & Graff, 2017), Compas (Brennan et al., 2009), Law School (Wightman, 2017), and the Health Heritage Prize (Kaggle, 2012) dataset. Below, we briefly introduce each of these datasets and discuss whether they contain personally identifiable information, if applicable. We preprocess Adult and Compas into categorical datasets by discretizing continuous features, keeping the other datasets as continuous. We drop rows and columns with missing values. For each dataset, we first split the data into training and test set, using the original splits wherever possible and a 80% / 20% split of the original dataset otherwise. We then further sample 20% of the training set to be used as validation set. Table 3 displays the dataset statistics for each of these splits. Finally, we drop uninformative features to facilitate density estimation. We show that the removal of these features does not significantly affect the predictive utility of the data in Table 4.

**Adult**

The Adult dataset, also known as Census Income dataset, was extracted from the 1994 Census database by Barry Becker and is provided by the UCI machine learning repository (Dua & Graff, 2017). It contains 14 attributes: age, workclass, fnlwgt, education, education-num, marital-status, occupation, relationship, race, sex, capital-gain, capital-loss, hours-per-week, and native-country. The prediction task is to determine whether a person makes over 50 000 US dollars per year. We consider sex as the protected attribute, and we discretize the dataset by keeping only the categorical columns relationship, workclass, marital-status, race, occupation, education-num, and education.

**Compas**

The Compas (Brennan et al., 2009) dataset was procured by ProPublica, and contains the criminal history, jail and prison time, demographics, and COMPAS risk scores for defendants from Broward County from 2012 and 2013. Through a public records request, ProPublica obtained two years worth of COMPAS scores (18 610 people) from the Broward County Sheriff's Office in Florida. This data was then augmented with public criminal records from the Broward County Clerk's Office website.

Table 4: Classification accuracy before and after removing uninformative features during preprocessing. For each dataset we train a multi-layer perceptron (MLP) with the same architecture on the original and preprocessed data and report the average accuracy with standard deviation for five different random seeds. We can observe that the accuracy only decreases slightly after preprocessing.

|  | ACCURACY (%) | |
|---|---|---|
|  | ORIGINAL | PREPROCESSED |
| ADULT | 85.0 ($\pm$ 0.001) | 84.4 ($\pm$ 0.001) |
| COMPAS | 65.3 ($\pm$ 0.003) | 65.0 ($\pm$ 0.003) |
| CRIME | 85.5 ($\pm$ 0.003) | 85.2 ($\pm$ 0.004) |
| HEALTH | 80.1 ($\pm$ 0.002) | 76.1 ($\pm$ 0.001) |
| LAW | 88.2 ($\pm$ 0.006) | 86.4 ($\pm$ 0.001) |

Furthermore, jail records were obtained from the Browards County Sheriff's Office, and public incarceration records were downloaded from the Florida Department of Corrections website. The task consists of predicting recidivsm within two years for all individuals. We only consider Caucasian and African-American individuals and use race as the protected attribute. We discretize the continuous features age, diff-custody, diff-jail, and priors-count, and we remove all other features except for sex, c-charge-degree, and v-score-text.

**Crime**

The Communities and Crime dataset combines socio-economic data from the 1990 US Census, law enforcement data from the 1990 US LEMAS survey, and crime data from the 1995 FBI UCR. It was created by Michael Redmond and is provided by the UCI machine learning repository (Dua & Graff, 2017). The dataset contains 128 attributes such as county, population, per capita income, and number of immigrants. The task consists of predicting whether the number of violent crimes per population for a given community is above or below the median. We consider race as the protected attribute, which we set to 1 if the percentage of white people divided by 5 is smaller than the percentage of black, asian, and hispanic individuals, and to 0 otherwise. We keep the following 6 features: racePctWhite, pctWInvInc, PctFam2Par, PctKids2Par, PctYoungKids2Par, PctKidsBornNeverMar.

**Health**

The Health dataset was created for the Heritage Health Prize (Kaggle, 2012) competition on Kaggle and contains medical records of over 55 000 patients. We consider the merged claims, drug count, lab count, and members sheets, which have a total of 18 attributes. The identity of individual patients and health care providers, as well as other individual identifiable information, has been removed from the datasets to protect the privacy of those involved and to comply with applicable law. We consider age as the protected attribute, which we binarize to patients above and below 60 years. The task is to predict the maximum Charlson Comordbidity Index, which predicts 10-year survival in patients with multiple comorbities. We drop all but the following features: DrugCount-total, DrugCount-months, no-Claims, no-Providers, PayDelay-total, PrimaryConditionGroup, Specialty, ProcedureGroup, and PlaceSvc. For the transfer learning experiments we follow Madras et al. (2018) and omit the primary condition group labels from the set of features and try to predict them from the latent representation without explicitly optimizing for the task.

**Law School**

The Law School (Wightman, 2017) dataset contains admissions data from 25 law schools over the 2005, 2006, and in some cases 2007 admission cycles, providing information on over 100 000 individual applications. The data was procured by Project SEAPHE and was cleaned to adhere to high standards of data privacy. Concretely, when the school, race, year, and gender information for enrolled students produced cells of fewer than five subjects, the cells were combined to minimize reidentification risk. The attributes are law school, year of fall term, LSAT score, undergraduate GPA,

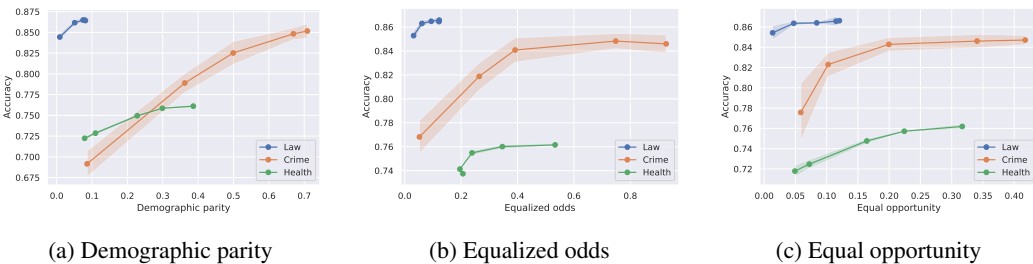

|  (a) Demographic parity | (b) Equalized odds | (c) Equal opportunity |

Figure 6: Tradeoff between accuracy and various fairness metrics (demographic parity, equalized odds, equal opportunity) when using Fair Normalizing Flows (FNF).

race, gender, and in-state residency. We consider race as the protected attribute, which we binarize to white and non-white. We remove all features but the LSAT score, undergraduate GPA, and the college to which the student applied (ordered by decreasing admission rate).

## B.2 TRAINING DETAILS

Our code is implemented in PyTorch (Paszke et al., 2019).

**Computing resources**   We run all experiments on a desktop PC using a single GeForce RTX 2080 Ti GPU and 16-core Intel(R) Core(TM) i9-9900K CPU @ 3.60GHz.

**Hyperparameters for main experiments**   For Crime we estimate input density using Gaussian Mixture Model (GMM) with 4 components for $a = 0$ and 2 components for $a = 1$. For Law we use GMM with 8 components for both groups. The Health dataset requires more complex density estimation so we use RealNVP (Dinh et al., 2016) with 4 blocks of 20 neurons each. For categorical datasets, Adult and Compas, we perform density estimation using MADE (Germain et al., 2015), which is represented using network of 2 hidden layers with 50 neurons.

We represent flow encoders using RealNVP with 4 blocks for Crime and Law, and 6 blocks for Health. Crime and Law use batch size 128, initial learning rate 0.01 and weight decay 0.0001, while Health uses batch size 256, initial learning rate 0.001 and weight decay 0. Training is performed using Adam (Kingma & Ba, 2015) optimizer. We use 60, 100, and 80 epochs for Crime, Law and Health, respectively. These parameters were chosen based on the performance on validation set.

For experiments in Fig. 4, we trained with the following 5 values for $\gamma$ for respective datasets: 0, 0.02, 0.1, 0.2, 0.9 for Crime, Adult and Compas, 0, 0.001, 0.02, 0.1, 0.9 for Law, 0, 0.05, 0.1, 0.5, 0.95 for Health. Training for 1 epoch takes around 1 second for Crime, 5 seconds for Law, and 30 seconds for Health.

## C   ADDITIONAL EXPERIMENTS

In this section we present additional experimental results.

**Compatibility with other fairness metrics**   In Section 6 we focused on presenting results on statistical distance, as it can bound various fairness metrics Madras et al. (2018). In constrast, here we provide more detailed experiments with three common group fairness metrics: demographic parity, equalized odds, and equality of opportunity. We demonstrate the tradeoff between these metrics and downstream accuracy in Fig. 6. We observe that FNF achieves high rates of demographic parity, equalized odds, and equality of opportunity with only small decreases in classification accuracy (similar to the results for the statistical distance showed in Fig. 4).

**Compatibility with different scalarization schemes**   Since fairness and accuracy are often competing objectives, they merit a treatment from multi-objective optimization. Here, we investigate the scalarization scheme proposed by Wei & Niethammer (2020) and

replace our objective $\gamma(\mathcal{L}_0 + \mathcal{L}_1) + (1 - \gamma)\mathcal{L}_{clf}$, obtained via convex scalarization, with the Chebyshev scalarization scheme $\max\{\gamma(\mathcal{L}_0 + \mathcal{L}_1), (1 - \gamma)\mathcal{L}_{clf}\}$, where we use the same normalization for $\mathcal{L}_0$, $\mathcal{L}_1$, and $\mathcal{L}_{clf}$ as Wei & Niethammer (2020).

We evaluate the schemes for a large range of $\gamma$ values on the Crime dataset and compute the Area Under the Curve (AUC) with the trapezoidal rule. The convex scalarization yields an AUC of 0.6036, whereas the Chebyshev scalarization attains an AUC of 0.6051. Moreover, we aggregate the results in Fig. 7. In general, we observe that the convex scalarization slightly outperforms the Chebyshev scalarization scheme. We believe that this is due to two reasons, (i) the Pareto curve is almost convex, which is why the convex scalarization performs well, and (ii) the stochasticity of gradient-based optimization. We consider more advanced multi-objective optimization methods Lin et al. (2019); Martínez et al. (2020) an interesting direction for future work.

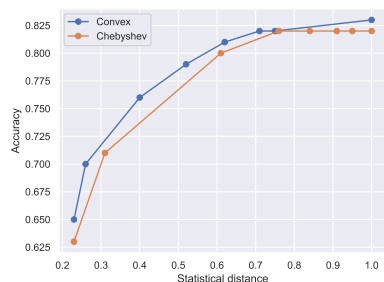

Figure 7: Different scalarization schemes.

**Compatibility with different priors** In the following experiment, we demonstrate that FNF is compatible with any differentiable estimate of $p_0$ and $p_1$. We consider Crime with the same setup as before, but with 3 different priors: a GMM with $k = 3$ components, an autoregressive prior, and a RealNVP flow (Dinh et al., 2016). For each of these priors, we train an encoder using FNF and a classifier on top of the learned representations. Fig. 8 shows the tradeoff between the statistical distance and accuracy for each of the priors. Based on these results, we can conclude that FNF achieves similar results for each of the priors, empirically demonstrating the flexibility of our approach.

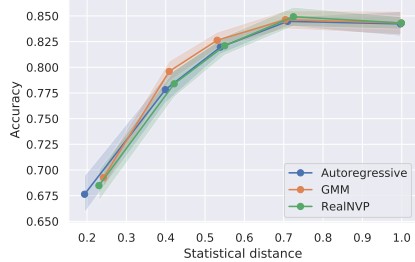

Figure 8: Different priors used with FNF.

**Interpreting the representations** We now show that the bijectivity of FNF enables interpretability analyses, an active area of research in fair representation learning (He et al., 2019; Kehrenberg et al., 2020; Wang et al., 2021). To that end, we consider the Crime dataset where for a community $x$: (i) race (non-white vs. white majority) is the sensitive attribute $a$, (ii) the percentage of whites (highly correlated with, but not entirely predictive of the sensitive attribute) is in the feature set, and (iii) the label $y$ strongly correlates with the sensitive attribute. We encode every community $x \sim p_a$ as $z = f_a(x)$ and compute the corresponding community with opposite sensitive attribute that is also mapped to $z$, i.e., $\tilde{x} = f_{(1-a)}^{-1}(z)$ (usually not in the dataset), yielding a matching between both distributions. We visualize the t-SNE (van der Maaten & Hinton, 2008) embeddings of this mapping for $\gamma \in \{0, 1\}$ in Fig. 9, where the dots are communities $x$ and the crosses are the corresponding $\tilde{x}$. We run k-means (MacQueen, 1967; Lloyd, 1982) on all points with $a = 1$ and show the matched clusters for points with $a = 0$ (e.g., red clusters are matched). For $\gamma = 0$, where FNF only minimizes the classification loss $\mathcal{L}_{clf}$, we observe a dichotomy between $x$ and $\tilde{x}$, since the encoder learns to match real points $x$ with high likelihood to points $\tilde{x}$ with low likelihood, yielding both high task utility (due to (iii)), but also high statistical distance (due to (ii)). For example, the red cluster has an average log-likelihood of $-4.3$ for $x$ and $-130.1$ for $\tilde{x}$. In contrast, for $\gamma = 1$ FNF minimizes only the KL-divergence losses $\mathcal{L}_0 + \mathcal{L}_1$, and thus learns to match points of roughly equal likelihood to the same latent point $z$ such that the optimal adversary can no longer recover the sensitive attribute. Accordingly, the red cluster has an average log-likelihood of $-2.8$ for $x$ and $-3.0$ for $\tilde{x}$.

**Tradeoff between accuracy and fairness** Zhao & Gordon (2019) proved that any classifier with perfect statistical distance necessarily has to sacrifice classification accuracy, where the exact tradeoff depends on the difference in base rates between the different sensitive groups. We confirm this statement, with an investigation of the tradeoff between accuracy and fairness for the Crime dataset, where we need to sacrifice a significant amount of accuracy in order to decrease the statistical distance (as can be observed in Fig. 4).

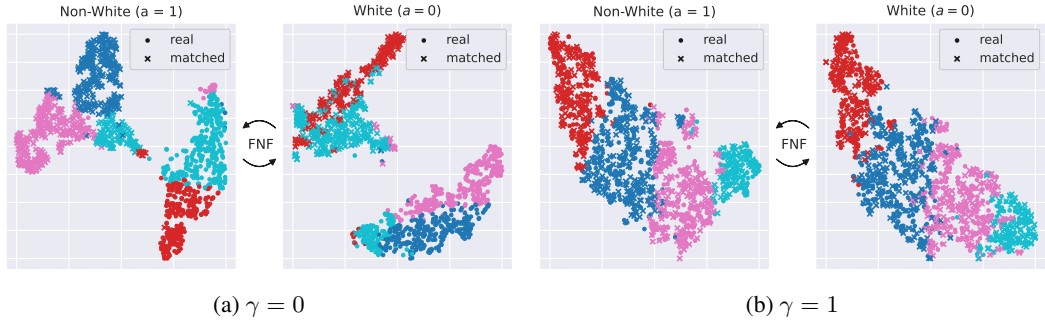

(a) $\gamma = 0$        (b) $\gamma = 1$

Figure 9: Visualizing k-means clusters on t-SNE embeddings of mappings between real points from the Crime dataset and their corresponding matches from the opposite attribute distribution.

For each pair of sensitive attribute $a$ (racial group) and label $y$ (whether the number of violent crimes is above the median) we report the probability $P(Y = y | A = a)$ in Table 5 (the probabilities for the other datasets can be found in Table 3). Clearly, Table 5 shows that the sensitive attribute $a$ and the task label $y$ of the Crime dataset are highly correlated: one racial group was much more likely to be reported for violent crimes than the other. This is, of course, a consequence of bias in the data (e.g., some neighborhoods tend to be policed more often so more crimes will be reported), as has been documented in various studies, e.g., see Brennan et al. (2009) for a study of the Compas dataset. Thus, in accordance with the result from Zhao & Gordon (2019), we need to sacrifice a lot of accuracy to achieve a small statistical distance on the Crime dataset.

Table 5: Statistics for the Crime train set (reproduced from Table 3).

|  | $y = 0$ | $y = 1$ |
|---|---|---|
| $a = 0$ | 0.29 | 0.71 |
| $a = 1$ | 0.82 | 0.18 |

