# OpenReview forum: "Fair Normalizing Flows"
_ICLR.cc/2022/Conference — ICLR 2022 Poster_

### Official Review · Reviewer_BBgE · 2021-10-27

**Correctness:** 4
**Technical Novelty And Significance:** 4
**Empirical Novelty And Significance:** 4
**Recommendation:** 6
**Confidence:** 4

**Details Of Ethics Concerns:**

No concerns.

**Main Review:**

Strengths

* Fair representation learning with theoretical guarantees for fairness is a significant contribution.
* The algorithm for training the encoders and minimizing the KL divergence is straightforward and practical.
* Even if the probability distribution of the input data is not known, density estimation can be used.
* The invertible encoders can be used for interpretation and transfer learning purposes.
* Experiments are extensive and show how FNF outperforms adversarial training baselines and suggests upper-bound performances of any adversary.

Weaknesses

* Most of the analyses focus on the fairness guarantee, but there is not much explanation on why the accuracy does not degrade much. The experiments are extensive enough (including the ones in the supplementary), but it is still not clear why FNF should have less accuracy tradeoffs than adversarial training baselines in general. I suggest the authors also show the Figure 4 results for adversarial training.
* In Section 2, Gupta et al. (2021) is mentioned as a relevant work, but is considered inferior due to its monotonic nature of bounds. However, it would be interesting to also see any experiments whether FNF actually outperforms this approach.
* In Page 6, how tight is the bound of the square root of KL divergence for statistical distance? FNF's performance seems to largely rely on this tightness.
* In Algorithm 1, there should be some discussion on when to stop the learning. That is, how can one set $N$?
* Minor comments
   * In Page 7, Figure 4 seems to appear too early. The description is in the next Page 8, which is confusing. In Page 8, Table 1 also appears a bit too early.
   * The experimental setting for Table 1 is not clear. Which datasets were used and were the results averaged?



**Summary Of The Paper:**

This paper proposes a representation learning technique for fairness called Fair Normalization Flows (FNF) with theoretical guarantees on the maximum-possible unfairness. The key idea is to train encoders that map input samples of two sensitive groups into a space where the two groups become indistinguishable in terms of statistic distance (=maximum mean discrepancy) without significantly sacrificing accuracy. FNF minimizes the KL divergence between the converted distributions of the groups, which is an upper bound of statistic distance. In comparison to previous approaches that use adversarial training, FNF does not have instability issues when solving a non-convex problem. In addition, FNF has interpretability and transfer learning functionalities, which other fair representation learning techniques lack. Experiments show that FNF ensures fairness while retaining high accuracy for various real datasets and outperforms other adversarial training baselines.

**Summary Of The Review:**

The paper makes significant contributions by proposing a fair representation learning technique with theoretical bounds. The paper can also be improved by better explaining the accuracy results and adding some more details of its techniques and experiments.

---

> ### Author Response · Authors · 2021-11-16
> **Response to Reviewer BBgE**
>
> Thank you for your thoughtful remarks, which we address below.
>
> **Can you show experiments demonstrating that FNF outperforms the approach by Gupta et al. (2021)?**
>
> The two approaches are not actually comparable as Gupta et al. (2021) focus on demographic parity and provide no way to estimate the statistical distance. In contrast, as shown in Figure 6 in the Appendix, we are not limited to demographic parity, and we can also bound equalized odds and equal opportunity.
>
> **How tight is the bound of the square root of the KL divergence for the statistical distance?**
>
> While we cannot explicitly compute the tightness of this bound, our experimental results indicate that the bound is quite tight as it allows us to train models with a very small statistical distance.
>
> **Why does the accuracy not degrade much compared to adversarial training?**
>
> The key reason is that methods based on adversarial training only protect against adversaries from the adversarial family $\mathcal{G}$, while FNF protects against all adversaries, as we train against the optimal adversary (from any family). As mentioned in the paper, this issue with adversarial training based methods is thoroughly explained and investigated in other works (Feng et al., 2019; Elazar & Goldberg, 2018; Xu et al., 2020; Gupta et al., 2021), so we believe it is not necessary to repeat the same experiments here, and Table 1 should be sufficient.
>
> **How many iterations of Algorithm 1 do you typically perform?**
>
> As discussed in Appendix B.2, we use 60, 100, and 80 epochs for Crime, Law, and Health,
> respectively. These parameters were chosen based on the performance on the validation set.
>
> **What is the experimental setting for Table 1?**
>
> The experiment was performed on the Adult dataset. We have now added this into the description of the experiment.

---

### Official Review · Reviewer_Vaqy · 2021-10-29

**Correctness:** 3
**Technical Novelty And Significance:** 2
**Empirical Novelty And Significance:** 2
**Recommendation:** 5
**Confidence:** 4

**Main Review:**

Strengths of the paper
+ The paper considers the relevant task of learning fair representations
+ The issues with learning fair representation via adversarial training are explained well

Weaknesses of the paper
+ Algorithm 1 is incomplete. No details are given about $\mathcal L_{clf}$. Shouldn't we be updating the parameters of the classifier $h$ in Algorithm 1?
+ There is no connection between Algorithm 1 and the various theoretical results given in the paper. You must show that minimizing $\mathcal L$ in Algorithm 1 ensures that $\Delta(p_{Z_0}, p_{Z_1})$ is small so that in turn the maximum adversarial accuracy is small. This is not obvious since $\mathcal L$ in Algorithm 1 also contains $\mathcal L_{clf}$.
+ The scalarization scheme employed in Algorithm 1's loss function $\mathcal L = \gamma (L_0 + L_1) + (1-\gamma) L_{clf}$ is well known to be the most naive, and ineffective scalarization technique in multi-objective optimization. A proper application of multi-objective optimization techniques will produce far more interesting trade-off curves than what's seen in Figure 4. Good references are "The Fairness-Accuracy Pareto Front" by Wei and Niethammer and "Pareto Multi-Task Learning" by Xin et al.
+I won't be the only one pointing this out, but here goes: a big weakness of the paper is the assumption that $p_0$ and $p_1$ are known. I do note that you reference several fairness papers at the end of Section 5 that have apparently "successfully" employed density estimation. But this isn't even my biggest concern. Again, I am afraid that too many uncontrolled approximations are made along the way. It is worth highlighting that $\mathcal L$ in Algorithm 1 does not even contain $\Delta(p_{Z_0}, p_{Z_1})$, but rather the symmetrized KL-Divergence between $Z_0$ and $Z_1$ which is simply not the same as the statistical distance between $Z_0$ and $Z_1$.

Minor questions
+ Why use the term statistical distance and not MMD?
+ According to the Background (Section 3), $y$ is binary. Consider replacing throughout the expression $E(y=h(x))$ with simply $P(y=h(x))$.
+ In the left panel of Figure 2, consider adding different symbols to illustrate the two classes $y=0$ and $y=1$. Also consider coloring the $(x_1,x_2)$ region according to the classifier $h(z)$ where $z$ is as given in Section 4.
+ In figures such as Fig 4a and 4b, consider adding the value of $\gamma$ above each point.

**Summary Of The Paper:**

Given input data $x$, target $y$, and a binary sensitive attribute $a$, this work undertakes the task of learning a representation of $x$, call it $z$, such that a classifier $h$ can perform well using $z$ while at the same time no adversary $g$ can predict $a$ well from $z$.

This is accomplished by minimizing, see Algorithm 1,
$$
\gamma \frac{1}{2} D_{KL}(p_{Z_0} || p_{Z_1}) + (1-\gamma) \mathcal L_{clf}.
$$
This operation is possible because 1) $p_0$ and $p_1$, the distributions of $x$ for $a=0$ and $a=1$, respectively, are presumed to be given or readily estimable, and 2) the latent $z$ is obtained by normalizing flows $f_0$ and $f_1$ applied to $p_0$ and $p_1$, respectively.

Supposedly this optimization problem results in latent representations $z$ that ensures the statistical distance/MMD $\Delta(p_{Z_0}, p_{Z_1})$ is small so that in turn the maximum adversarial accuracy is small.

**Summary Of The Review:**

Much fuss is made about controlling the statistical distance between the latent representations $z_0$ and $z_1$. This is justified because controlling the statistical distance amounts to controlling the maximum adversarial accuracy and additionally gives guarantees on demographic parity and equalised odds for any downstream classifier $h$ that uses $z$.

However what is actually implemented in Algorithm 1 has no clear connection to controlling the statistical distance between $z_0$ and $z_1$. This feels like a sleight of hand to me. Certainly none of the theory supports Algorithm 1's ability to control the statistical distance. Furthermore no careful thinking is given to the tradeoff value of $\gamma$ in Algorithm 1.

---

> ### Author Response · Authors · 2021-11-16
> **Response to Reviewer Vaqy**
>
> Thank you for your constructive feedback, which we address below.
>
> **Why do you only use the convex scalarization scheme for your multi-objective optimization problem?**
>
> As mentioned in the paragraph “Training flow-based encoders” on Page 6, in Appendix C, we *already* compared the convex scalarization technique with the (normalized) Chebyshev scalarization scheme (Wei and Niethammer, 2020). Our results indicated that both schemes produce similar results.
>
> **How can you justify the assumption of knowing $p_0$ and $p_1$?**
>
> As reviewer `R-BBgE` stated, our guarantees also hold with estimated densities (see Theorem 5.3), and Figure 5 demonstrated that the guarantees obtained via density estimation transfer to real-world models.
>
> **Do you need to prove that minimizing the loss of Algorithm 1 minimizes adversarial accuracy?**
>
> No, this is not necessary. We could use any training algorithm and measure statistical distance in the end to obtain the bound on adversarial accuracy. If the training algorithm did not consider fairness, that bound would be loose (but still correct). In that sense, our Algorithm 1 provides one way which works well in practice, as our experimental results indicate.
>
> **Are you also updating the parameters of the classifier $h$ in Algorithm 1?**
>
> Yes, the parameters of h are updated.
>
> **Why do you use the term statistical distance and not MMD?**
>
> In accordance with prior work (Madras et al., 2018), we use the term statistical distance to denote the supremum of the MMD between $\mathcal{Z}_0$ and $\mathcal{Z}_1$.

---

> > ### Comment · Reviewer_Vaqy · 2021-11-17
> > **thoughts**
> >
> > 1. Apologies for missing your material in Appendix C.
> > 2. You answered "Yes, the parameters of h are updated." But this is not apparent from the way Algorithm 1 is written. Shouldn't Algorithm 1 be more clear about $h$ and the associated $\mathcal L_{clf}$?
> > 3. Thank you for the proof justifying symmetrized KL in Algorithm 1 as a differentiable proxy for statistical distance via Pinsker's in Appendix A.1. However there still remains a disconnect between your implementation and your theory. You are not merely minimizing symmetrized KL ($\mathcal L_0 + \mathcal L_1$) in Algorithm 1. Rather in Algorithm 1, you are minimizing $\gamma$*symmetrized KL + $(1-\gamma) \mathcal L_{clf}$. When you minimize this combined loss, the type of guarantee you can get on statistical distance crucially depends on $\gamma$. When I look at Figure 5, which is a nice result, I wonder what is the $\gamma$ that produced this nice result. I suppose $\gamma$ really close to $1$? Well then we sacrifice classification accuracy. Is that worth it?
> > 4. Table 2 partly confirms my concerns. To guarantee a small statistical distance $\Delta$, we lose quite a bit of accuracy. (It's curious that out of all the datasets you explore, Crime seems to have the most pronounced tradeoffs between statistical distance and classification accuracy.)
> > 5. The first sentence of the conclusion states "We introduced Fair Normalizing Flows (FNF), a new method for learning representations which ensure that no adversary can predict sensitive attributes." There are various sentences throughout the manuscript that express the same sentiment. I would suggest tempering such statements with the sobering truth that to reduce the maximum adversarial accuracy, we might also have to lose on classification accuracy.
> >
> > I've increased the score to reflect my updated views on using the symmetrized KL as a surrogate for statistical distance.

---

> > > ### Author Response · Authors · 2021-11-19
> > > **Response**
> > >
> > > Thank you for your quick reply and for updating your score.
> > >
> > >
> > > We have now updated the paper by incorporating your suggestions:
> > >
> > > * We updated Algorithm 1 to be more explicit about $h$.
> > > * In Appendix C, we clarified the tradeoff between accuracy and statistical distance with the investigation of the tradeoff for the Crime dataset.
> > > * Throughout the paper, we added statements that classification accuracy has to be sacrificed to reduce the statistical distance.
> > >
> > > Below we address your questions. If you are satisfied with our response, we kindly ask you to consider increasing your score, and otherwise, please let us know if you have any further concerns.
> > >
> > > **Can Algorithm 1 be more explicit about the fact that $h$ is updated?**
> > >
> > > Yes, we have now updated Algorithm 1 to reflect the fact that $h$ is also updated during the training.
> > >
> > > **What are the $\gamma$ values that produced results in Figure 5? Is $\gamma$ close to 1? Is sacrificing classification accuracy worth it?**
> > >
> > >
> > > Note that Figure 5 uses the same models as Figure 4, as mentioned in the text, and different $\gamma$ values used to obtain these models were listed in Appendix B.2 (unfortunately, there was not enough space to put these values in the main text). We have also updated the text in B.2 to say that the $\gamma$ parameters for Adult and Compas were the same as Crime, as this was not explicitly written before. For example, for Health we use the following $\gamma$ values: $0, 0.05, 0.1, 0.5, 0.95$. As we can observe in Figure 4, even very small $\gamma = 0.05$ significantly decreases statistical distance by $0.54$, compared to the case when $\gamma = 0$ (which only minimizes classification loss), while sacrificing only $0.003$ accuracy.
> > >
> > > Whether sacrificing accuracy for fairness is worth it certainly depends on the individual user, as they might have different preferences for how much accuracy they are willing to sacrifice. Thus, they would have the option to choose the $\gamma$ value according to their preference.
> > >
> > > **Do we need to lose quite a bit of accuracy to guarantee a small statistical distance?
> > > Why does Crime have the worst tradeoff?**
> > >
> > > There has been a lot of work investigating the tradeoff between fairness and accuracy [1, 2, 3]. For example, Zhao & Gordon [3] prove that any classifier with perfect statistical distance necessarily has to sacrifice accuracy, where the exact tradeoff depends on the difference in base rates between different groups. Intuitively, as soon as there exists at least some correlation between the sensitive attribute and the label, there cannot exist a classifier that is both fair and accurate, meaning that each user has to decide how much accuracy they are willing to sacrifice to increase fairness.
> > >
> > > This is also a reason why we observe such a drastic tradeoff on Crime in our experiments. If we measure for each pair of sensitive attribute $a$ (racial group) and label $y$ (whether the number of violent crimes is above the median) the probability $P(Y=y \mid A=a)$, we obtain the following table (note that these rates can be found for all the datasets in Table 3):
> > >
> > > |           | $y=0$| $y=1$|
> > > | --------- |:----:| ----:|
> > > | $a = 0$   | $0.29$ | $0.71$ |
> > > | $a = 1$   | $0.82$ | $0.18$ |
> > >
> > > Clearly, the table shows that the sensitive attribute $a$ and the task label $y$ are highly correlated: one racial group was much more likely to be reported for violent crimes than the other. This is, of course, a consequence of bias in the data (e.g., some neighborhoods tend to be policed more often, so more crimes will be reported), as has been documented in various studies, e.g., see [4] for a study of the Compas dataset. In that sense, the work of Zhao & Gordon tells us that we should expect to sacrifice a lot of accuracy to achieve a small statistical distance on the Crime dataset (regardless of the representation learning method), which is precisely what happens in our experiments, as you have pointed out.
> > >
> > > We have now added this discussion to Appendix C.
> > >
> > > **Can you reformulate your claims to reflect the fact that your method might reduce classification accuracy to reduce the maximum adversarial accuracy?**
> > >
> > > Yes, in the updated version, we have reformulated our claims throughout our paper (introduction/background/conclusion) to caution practitioners that our method might decrease the classification accuracy due to the fairness-accuracy tradeoff discussed above.
> > >
> > > **References**
> > >
> > > [1] Zemel, Rich, et al. "Learning fair representations." International conference on machine learning. PMLR, 2013.
> > >
> > > [2] Hardt, Moritz, Eric Price, and Nati Srebro. "Equality of opportunity in supervised learning." Advances in neural information processing systems 29 (2016): 3315-3323.
> > >
> > > [3] Zhao, Han, and Geoff Gordon. "Inherent tradeoffs in learning fair representations." Advances in neural information processing systems 32 (2019): 15675-15685.
> > >
> > > [4] https://www.propublica.org/article/machine-bias-risk-assessments-in-criminal-sentencing

---

> > > > ### Comment · Reviewer_Vaqy · 2021-11-21
> > > > **The relevance of your objective function**
> > > >
> > > > Thank you for the many clarifications. My main concern still remains however. I've said in one way or another in all my comments that there is a gap in what you formulate as your objective function
> > > > $$
> > > > \gamma \frac{1}{2} D_{KL}(p_{Z_0} || p_{Z_1}) + (1-\gamma) \mathcal L_{clf}
> > > > $$
> > > > and all the claims made in your theoretical results.
> > > >
> > > > I was interested to see the following Q+A with reviewer yCUe.
> > > >
> > > > *Do you guarantee that you can provide a good (accurate/probable) estimate of statistical distance, and not that it will be low after training? Could you make it more clear in the abstract/intro?*
> > > >
> > > > *Yes, this is precisely the correct understanding. Essentially, we have a training procedure in Algorithm 1 that we use to train encoders that, in practice, have a small statistical distance (as can be seen from experimental evaluation). Once training is finished, we can provide an estimate of the statistical distance that holds with high probability. Naturally, proving that training converges to the model with low statistical distance would be significantly more difficult (e.g., one would have to argue about the convergence of SGD for normalizing flows). Though we do not have that much space right now to elaborate in detail, we have slightly modified the introduction to make this point more clear (if paper is accepted we would have an extra page which we would use to clarify this further).*
> > > >
> > > > I would certainly not dare to ask that you prove convergence of SGD for normalizing flows. I am merely interested in the properties of the *theoretical* minimizer of your objective function. Does it return a classifier with guaranteed low statistical distance, at least for certain values of $\gamma$? It seems to me that unless $\gamma$ is very close to 1, your lemmas/theorems do not apply, i.e. you cannot guarantee low statistical distance for the *theoretical* minimiser of the objective function for most values of $\gamma$. If I am correct, then I do not understand the contributions of the paper. I don't need your method, after all, to estimate the statistical distance. I can just estimate it.

---

> > > > > ### Author Response · Authors · 2021-11-21
> > > > > **Response**
> > > > >
> > > > > Thank you for your response. We hope to clarify your questions regarding our method below and are happy to address any further questions.
> > > > >
> > > > > **I do not understand your contributions. Can’t I just estimate the statistical distance myself?**
> > > > >
> > > > > No, prior work employing standard models (e.g., feedforward neural networks) cannot estimate the statistical distance of the learned representations since these models can map many inputs $x$ to the same latent point $z$, thus preventing the application of the inverse function theorem to compute the latent densities $p_{Z_0}(z)$ and $p_{Z_1}(z)$ (as explained after Lemma 5.1).
> > > > >
> > > > >
> > > > > To achieve this, we need the following ingredients (which are all our contributions):
> > > > >
> > > > > * Encoder architecture based on normalizing flows that enables the estimation of the statistical distance in the latent space.
> > > > > * Training procedure (our Algorithm 1) for the normalizing flow encoders (note that we do not need any guarantees for this part as explained in the next question).
> > > > > * Given trained encoders, we need Lemmas 5.1 - 5.4, Theorem 5.5 to provide *high probability* estimate of statistical distance that consequently provides a *provable* upper bound on the standard fairness metrics.
> > > > >
> > > > > It would be great if the reviewer could further clarify (more formally) how they plan to estimate statistical distance of learned representations without using our contributions, or point to some prior work which they believe can achieve this, as we are not aware of any such approach.
> > > > >
> > > > > **Is it necessary to show that the theoretical minimizer of your objective guarantees low statistical distance?**
> > > > >
> > > > > No, this is neither necessary nor practically useful. Note that, even if you were able to show this, it would still have no practical utility for the user of this algorithm, unless you could prove that encoder with normalizing flows converges to these theoretically optimal parameters when trained using SGD. What is more useful, and what we do in this work, is to provide *empirical* training procedure (Algorithm 1), and then once training is finished provide *provable* estimate of statistical distance to the user (by applying Lemmas 5.1 - 5.4.). We provide an extensive experimental investigation showing that our method reduces the statistical distance across various datasets and for a large range of $\gamma$ values (from $\gamma = 0.001$ to $\gamma = 0.95$), as also shown in the response below.
> > > > >
> > > > > **Do your lemmas/theorems only apply for $\gamma$ very close to 1?**
> > > > >
> > > > > No, for example, Figure 4 shows that on the Law School dataset, our approach reduces the statistical distance from $\Delta = 0.644$ for $\gamma = 0$ to $\Delta = 0.152$ for $\gamma = 0.1$ (which we do not consider very close to $1$). Importantly, this decreases the maximum adversarial accuracy from $\frac{1 + 0.644}{2} = 0.822$ to $\frac{1 + 0.152}{2} = 0.576$ (with $0.5$ being random accuracy). Simultaneously, the accuracy of the classifier is essentially unaffected ($0.864$ for $\gamma = 0$ vs. $0.862$ for $\gamma = 0.1$).
> > > > > These numbers are estimates that hold with high probability, obtained using our lemmas/theorems, showing that they also work when $\gamma$ is not close to 1.
> > > > > We consider this convincing proof that our method can *significantly* reduce the statistical distance while leaving the accuracy roughly unaffected for $\gamma$ values that are far from being close to $1$.

---

> > > > > > ### Comment · Reviewer_Vaqy · 2021-11-30
> > > > > > **thank you**
> > > > > >
> > > > > > *It would be great if the reviewer could further clarify (more formally) how they plan to estimate statistical distance of learned representations without using our contributions, or point to some prior work which they believe can achieve this, as we are not aware of any such approach.*
> > > > > >
> > > > > > So if someone gives me learned representations $z_0^i$ and $z_1^i$, I could learn KDEs for $p_{Z_0}$ and $p_{Z_1}$ based on these samples. Now you may say, hold on, KDEs are terrible in high dimensions. I agree. But that's exactly the problem with Theorem 5.5. You push all the responsibility onto good density estimates $p_0$ and $p_1$ and these are also high dimensional and difficult to estimate.
> > > > > >
> > > > > > *Is it necessary to show that the theoretical minimizer of your objective guarantees low statistical distance?...No, this is neither necessary nor practically useful."*
> > > > > >
> > > > > > Alright, to each their own.
> > > > > >
> > > > > > *My question "Do your lemmas/theorems only apply for  very close to 1?"*
> > > > > >
> > > > > > My question was of a theoretical nature. You answered it by pointing to a few empirical examples. But it's okay.

---

> > > > > > > ### Author Response · Authors · 2021-11-30
> > > > > > > **Response**
> > > > > > >
> > > > > > > Thank you for your response. We address your concerns below.
> > > > > > >
> > > > > > > **Alright, to each their own.**
> > > > > > >
> > > > > > > We would appreciate it if you could provide reasons for why you disagree with our response and a more constructive feedback would certainly help us improve our paper.
> > > > > > >
> > > > > > > **Can I solve the problem by assuming that someone gives me learned representations, and I only estimate $p_{Z_0}$ and $p_{Z_1}$?**
> > > > > > >
> > > > > > > No. The problem is **precisely** that you are not given these representations, and instead you have to provide: 1) training method to learn the representations, 2) a method to estimate the statistical distance of the learned representations after the training is finished. What you are suggesting can be an approach to tackle the second step, but you are disregarding the first step. Note that using approaches from prior work for the first step does not work well as demonstrated in our Table 1 and plethora of other recent papers (Moyer et al., 2018; Elazar & Goldberg, 2018; Feng et al., 2019; Xu et al., 2020; Gupta et al., 2021; Song & Shmatikov, 2020). Our key contribution is to use normalizing flows for both the first step (training with Algorithm 1), and the second estimation step (our lemmas and theorems), which allows us to achieve better experimental results than prior work (Table 1) while providing some theoretical guarantees.
> > > > > > >
> > > > > > > **Why is estimating $p_0$ and $p_1$ better than estimating $p_{Z_0}$ and $p_{Z_1}$?**
> > > > > > >
> > > > > > > Note that $p_0$ and $p_1$ correspond to the low-dimensional distribution of interpretable features such as age or race, which are often estimated in the fairness literature. Estimating $p_{Z_0}$ and $p_{Z_1}$, which are distributions of non-interpretable features learned by a neural network, would be a more difficult problem. Just to give one technical reason why what you are suggesting is more difficult: guarantees obtained using KDE rely on knowing the bounds of the data distribution - if we are estimating age we know that this is a number between 0 and 120, but what can we say about the range of some latent variable learned by a deep neural network?
> > > > > > >
> > > > > > >
> > > > > > > Finally, it is important to point out that estimating original distribution is more interpretable and transparent, making it more likely to satisfy fairness regulations [1].
> > > > > > >
> > > > > > > [1] https://www.ftc.gov/news-events/blogs/business-blog/2021/04/aiming-truth-fairness-equity-your-companys-use-ai
> > > > > > >
> > > > > > > **Why do you point out a few empirical examples to show that your theorems apply?**
> > > > > > >
> > > > > > > Your question was asking whether our theorems apply to the cases when $\gamma$ is not close to 1. The reply said that theorems indeed apply, and they yield good results. We believe it is not fair to disregard our empirical results, as an important goal is to create a method that can be successfully applied in the real world.

---

> > > > > ### Comment · Reviewer_yCUe · 2021-11-26
> > > > > **Highlight on contributions.**
> > > > >
> > > > > Whilst there is not a proof regarding the tightness of the surrogate or formally examiniing the gap of the minimiser I think its very important to highlight that the theoretical gaurantees arising from the use of normalising flows in this work are not related to the surrogate/algorithm but are related to what is done after training.
> > > > >
> > > > > That is the main theoretical contribution is that this work allows us to inspect after training "how fair" our learned model is by being able to compute a closed form estimate of the statistical distance with high probability. I think this on its own is a very strong contribution as it allows practitionaires to examnise how "fair" their models are relative to this distance regadless of the algoirthm used to train (as long as they use flows / inverse transform theorem holds).
> > > > >
> > > > > Just wanted to highlight that this is the main contribution (the guarantees on estimating the MMD after training) which feels strong enough, the algorithm/surrogate itself is well motivated conceptually and whilst it may not be fully bulletproof in its theoretical analysis its not the first nor the last ML algorithm to lack such technical completeness and many of these methods have become ubiquotous and useful, sometimes with the more thorough analysis arising in later publications.
> > > > >
> > > > > Thought I would add to the discussion that their rebuttal is strong (clearly a lot of time put in) and it might be worth re-reading/visiting these final points before deciding on the final score.
> > > > >
> > > > > P.S. Quick note/clarifier: I also did not intend/date to ask the authors to analysis the result of SGD on a non-convex objectives, similar to you I was curious of what the theoretical miinimiser of the surrogate may be over some suitable space and wether certain types of flows/distributions/assumptions could make this question easier.  Whilst this would be a strong/nice add one I do think it would be quite a bit of work (does not come across as incredibly obvious/trivial). One plus side is this paper is opening the door for such questions which could be a promising direction for this subfield.
> > > > >
> > > > > Kind regards,
> > > > > Reviewer yCUe

---

> > > > > > ### Comment · Reviewer_Vaqy · 2021-11-30
> > > > > > **main theoretical contribution**
> > > > > >
> > > > > > Apologies for my delay in responding to all the interesting comments of late. Thanks to Reviewer yCUe for helping further articulate the contributions:
> > > > > >
> > > > > > *...its very important to highlight that the theoretical gaurantees arising from the use of normalising flows...*
> > > > > >
> > > > > > *...the main theoretical contribution is that this work allows us to inspect after training "how fair" our learned model is by being able to compute a closed form estimate of the statistical distance with high probability.*
> > > > > >
> > > > > > To a statistician, the present work does not actually demonstrate these contributions at all. Let me explain.
> > > > > >
> > > > > > 1. Does the work allows us to compute a closed form estimate of the statistical distance?
> > > > > >
> > > > > > Yes it does. But it is a simple tautological fact that once you have closed form expression for $p_{Z_0}$ and $p_{Z_1}$, you can compute a closed form expression of the statistical distance $\Delta(p_{Z_0}, p_{Z_1})$. It is also a simple tautological fact that once you have closed form expression of the estimates $\hat p_{z_0}$ and $\hat p_{z_1}$, you also have a closed form estimate of the statistical distance, $\Delta(\hat p_{Z_0}, \hat p_{Z_1})$.
> > > > > >
> > > > > > 2. We can get a closed form estimate of the statistical distance with high probability.
> > > > > >
> > > > > > It is true that Lemma 5.2 combined with Theorem 5.5 gives you this. However Theorem 5.5 is completely vacuous at the moment as it shifts all the responsibilities to estimating $p_0$ and $p_1$ well. I must say that it is unusual to see the crux of NF effectiveness resting on the source distributions $p_0$ and $p_1$. Why does Theorem 5.5 not consider the uncertainty in $f_0$ and $f_1$? Theorem 5.5 seems to take for granted that we can be given the true $f_0$ and $f_1$ and that all we have to figure out is the source distribution $p_0$ and $p_1$ in NF. This is very strange to me.
> > > > > >
> > > > > >
> > > > > > 3. There's something special about using normalizing flows here.
> > > > > >
> > > > > > Modeling the densities of $Z_0$ and $Z_1$ using normalizing flows gives rise to closed form expressions of $p_{Z_0}$ and $p_{Z_1}$. That is the *only* way in which normalizing flows are used theoretically in this paper. I could also model $Z_0$ and $Z_1$ using KDEs or parametric distributions (not that I'm advocating this) and still get closed-form expressions for $p_{Z_0}$ and $p_{Z_1}$. I would only agree that NFs are essential to the theoretical discussion *if* it can be shown that the proposed methodology actually satisfies the assumptions of Theorem 5.5.

---

> > > > > > > ### Comment · Reviewer_yCUe · 2021-11-30
> > > > > > > **main theoretical contribution**
> > > > > > >
> > > > > > > Thank you reviewer  Vaqy for the detailed clarification, it definitely adds very useful points to the discussion.
> > > > > > >
> > > > > > > > To a statistician, ..
> > > > > > >
> > > > > > > Moderately subjective language here ... and potentially making uncecesary assumptions and highlights about other reviewers backgrounds I think this tone/remark is unnecesary and could have been better phrased (e.g. from a statistical viewpoint).  My main point was :
> > > > > > >
> > > > > > > > 2. We can get a closed form estimate of the statistical distance with high probability.
> > > > > > >
> > > > > > > Which you add some valid points. There is a non practical nature to it in as it shifts the responsabilities to density estimation I agree this is not ideal. I disagree the use of the term vacous is accurate here ? maybe coloquialy there is a notion of "emptyness" due to just shifting the problem but its not particularly detailed. That said the statement is still true and is taking a step in this direction.

---

### Official Review · Reviewer_yCUe · 2021-11-02

**Correctness:** 3
**Technical Novelty And Significance:** 2
**Empirical Novelty And Significance:** 3
**Recommendation:** 8
**Confidence:** 4

**Details Of Ethics Concerns:**

The authors must be more explicit about the nature of the theoretical guarantees from the very beginning in the manuscript. At the moment the way the introduction and the abstract are worded make a first glance of the paper conclude that the statistical distance is guaranteed/likely to be small when using the approach proposed by the authors. However, this is far from what is in fact claimed by the more precise theorems which is that the empirically estimated statistical distance is guaranteed to be an accurate estimator of the true statistical distance. It's extremely important to make sure the claims are clear, especially since this work is on ethics and may be used in critical applications where a misguided reader may believe that the proposed approach is guaranteed (in probability) to be fair. When in fact the user should be estimating the statistical distance themselves in order to assess the level of fairness. Finally, the audience of this work may be less technical/more applied thus the guarantees and claims should be made clear and accessible to them prior to presenting the technical results.

**Main Review:**


General Suggestion:

It feels like the type of estimation results you are trying to obtain are very similar to prior works in non-vacuous generalisation bounds for DL. I think the authors work would benefit a lot from reading prior works done for DL/Generalisation such as [1,2] and use these to obtain a correct and computable alternative to that of theorem 5.1. Which as it stands is incorrect. I would be happy to modify my score subject to obtaining a valid result here and recomputing the estimate of the statistical distance across all experiments.

Questions to the Authors:
1. Where do you use $p(z)$ explicitly in your proofs/ algorithms ? There seems to be a big selling point on this; however besides theorem 5.3 I don't see a need for explicitly needing $p(z)$ , most of the other two claims 5.1 and 5.2 just require its existence but not its closed form.
2. As I mentioned the theory seems to be mostly estimation guarantees, you don't seem to be making a theoretical claim on bounding the fairness metrics themselves. Please clarify this.

Detailed Comments/Corrections:

1. Aesthetic: Maybe it's common in this community but using mathcal on a capital letter typically denotes a set\space (for example a set of functions) it's distracting\unhelpful to see it being used for a distribution.
2. Missing a $dz$ in Eq 4, \mu is a function not a measure. I would also suggest using f or g rather than \mu as in [3], since mu is incredibly commonly used for mesures, again this makes Eq 4 distracting/confusing at a glance.
3. Page 5: “However, for such models, given densities $p_0(x)$ and $p_1(x)$ over the input space, it is intractable to compute the densities $p_{Z0} (z)$ and $p_{Z1} (z)$ in the latent space as many inputs $x$ can be mapped to the same latent $z$”  . It could help the reader if you could be more explicit here, it seems the claim is that you can't carry out a change of variables to compute $p(z)$ because the inverse function theorem does not hold for most NN encoders and the Jacobian is not defined.
4. The Pinsker inequality bounds total variation not MMD. Please revisit MMD literature and use the appropriate bound / connections to KL/OT etc. If you can correct this (or correct me) I am happy to take this point back. Atm it really questions the KL motivation in Algorithm 1, you yet haven't shown that you are minimising a surrogate of the statistical distance. Additionally please translate this into a remark or a lemma so it stands out, as you can see from some of the reviews its easy to miss.
5. Why train a symmetrised KL while Pinsker's inequality is regarding KL ? there's no explanation for this when it is introduced. This needs justification why a symmetrised KL?.
6. Second instance of $\mu=1$ is a typo on the discussion of optimal adversary at the end of page 1 should be mu=0.
7. Lemma 5.1 : If we assumed the claim was right it would be a very vanilla concentration / CLT like type argument simply arguing that your empirical estimator of the statistical distance has nice convergence properties. This does not rely on flows and it doesn't give any theoretical arguments on how the authors approach results in a small adversarial accuracy. It does allow for empirical arguments of the adversarial accuracy to be made in probability but the authors would have to provide a delta column in all their experiments + estimated confidence bounds on their delta error too, which they haven't, this would give theoretically backed empirical estimators of the adversarial accuracy which would be a good contribution on its own , however this is currently lacking..
8. Lemma 5.1 should be written in a form similar to this for all  $\delta>0, P\left(\mathrm{err } < \frac{1}{\sqrt{n}} * \sqrt{\ln(2/\delta)} \right) \geq 1 - \delta$ so that it is clear to the reader what type of convergence rate you are showing. See theorem 1.1 in [2] for example.
9. Proof of Lemma 5.1 is abusing notation interchanging densities and measures (distributions). Again I would like to highlight that I am not sure what purpose the calligraphic Z notation has.
10. The proof for 5.1 has a misstep. Detail on the misstep : After applying the triangle inequality you distribute the sup over the 3 terms. You then use the definition of MMD and substitute the sup/expression in the middle term. However you also remove the sups from the 2 other terms leaving $\mu$ completely undefined. This does not even compile / type check as a proof. To apply a Hoeffding type bound on a sup you need to have the size of the hypothesis space and apply it via a union bound, as your inputs are real and your hypothesis space infinite this does not work so you go into VC-dimension type of bounds which are vacuous for neural nets. In the end what you need here is a PAC Bayes (e.g. [1,2] ) style of bound which has significant interpretation. In short you cannot apply Hoeffding's inequality over a sup like this. You could rectify this but it will be a different theorem. I think this could be done within a week so I would be happy to rectify my score if this + points 4 and 7 are rectified.
11. Looking at the proofs for Lemmas 5.1 and 5.3 I am really unconvinced that the flows are a necessary part of the proof ? All you require for these is to assume that the p(z) densities exist but at no point are flow properties used. In short the two provable guarantees you have provided apply to normal NN encoders too, the difference being that using flows will allow computing these statistical distances ?
12. The notation  $(..)dx = (..)dz$ is a bad heuristic and prone to errors please rectify this by carrying out the change of variables on $\int |p(x) - \hat{p}(x) | dx$ explicitly  pushing the jacobian determinant into the abs and applying the definition of p(z). Whilst the step/result is correct its very prone to mistake using heuristics with infinitesimals (imagine if we were looking at TV between $p_{N_1}(z)$ and $p_{N_0}(z)$ we wouldn't be able to apply the same heuristic here). Please correct this.
13. Lemma 5.2 seems to be used to motivate the weighting with gamma in Algorithm 1. That is: minimising the statistical distance (which is minimised surrogately so by minimising sym-KL) can lead to this “shuffling” which in turn will decrease classification accuracy and thus we have this gamma-tradeoff heuristic. I feel this lacks clarity. It's important to emphasize that this has nothing to do with the “provable guarantees” . I would argue it's more of a remark/observation that informs/motivates a sound heuristic. Please clarify.



[1] Dziugaite, G.K. and Roy, D.M., 2017. Computing nonvacuous generalization bounds for deep (stochastic) neural networks with many more parameters than training data. arXiv preprint arXiv:1703.11008.

[2] Dziugaite, G.K. and Roy, D.M., 2018. Data-dependent PAC-Bayes priors via differential privacy. arXiv preprint arXiv:1802.09583.

[3] Gretton, A., Borgwardt, K., Rasch, M., Schölkopf, B. and Smola, A., 2006. A kernel method for the two-sample-problem. Advances in neural information processing systems, 19, pp.513-520.



**Summary Of The Paper:**

This work proposes using normalising flows to obtain a latent representation of the data that is trained to minimise a surrogate of an MMD [3] based statistical distance used to measure adversarial accuracy when classifying protected features. All this whilst preserving the desired properties of the original features. The authors motivate using normalising flows arguing it allows to compute an empirical estimate of the aforementioned statistical distance which they claim has strong convergence guarantees.

**Summary Of The Review:**

Pros:
1. The goal of this work is very well motivated and much needed.
2. The authors provide a very nice illustrative example clarifying and motivating the need for these methods.
3. In general the authors do aim to motivate every algorithmic choice with some backing from theory which is a very nice structure/writing style.
4. Overall I do think the general direction the authors aimed to go in is very exciting and has a lot of promise. However in this current iteration it is lacking technical strength (both experimentally and theoretically).

Cons:

1.Unfortunately much of the theory the authors use to motivate design choices seem to have mistakes (Proof of Theorem 5.1 is incorrect, the usage of Pinsker’s inequality also seems wrong)

2. Because of the aforementioned error with Pinsker’s inequality  (it bounds TV not MMD) it's not clear if the KL based objective (as a surrogate to the statistical distance) is well motivated.

3. Not very safe heuristics are used in proofs (e.g. equating infinitesimals) ; this is prone to mistakes.

4. The use of normalising flows seems to be only used in theorem 5.3 ?

5. Finally the claimed theoretical results are “estimation guarantees” the way the introduction and abstract are worded seem to indicate that the theoretical results guarantee fairness for a given family of models, that is the way this work is selling itself seems to claim a bound on the true statistical distance. However that is far from the case, instead this work tries to prove that said distance can be estimated “very well” empirically. Thus all experiments in this work should be estimating this quantity as any success statements regarding fairness rely on such empirical estimates and there are really no theoretical guarantees on the fairness aspect of the method itself, it's more like there are theoretical guarantees on the evaluation of the method.

---

> ### Author Response · Authors · 2021-11-16
> **Response to Reviewer yCUe (Part 1)**
>
> Thank you for your detailed comments, positive feedback, and interesting questions. Below we address your concerns, which we will incorporate into the final version of our work. Due to the character limit, we split the response in two comments.
>
> **Question 1: Do you need to use $p(z)$ explicitly in your proofs and algorithms?**
>
> Yes, the density $p(z)$ is crucial in defining the optimal adversary $\mu^*$. More specifically, as described after Equation 4, it either assigns $\mu^*(z) = 1$ if and only if $p_{Z_0}(z) > p_{Z_1}(z)$ or it assigns $\mu^*(z) = 1$ if and only if $p_{Z_0}(z) < p_{Z_1}(z)$. Without being explicitly able to compute this optimal adversary, we cannot evaluate statistical distance (e.g., we would not be able to apply Lemma 5.1).
>
> **Question 2: Do you also make a theoretical claim on the fairness metrics?**
>
> Yes, Madras et al. (2018) have already proved that the statistical distance bounds various fairness metrics such as demographic parity or equalized odds. Thus, we can entirely focus on minimizing statistical distance. Figure 6 in the Appendix shows more detailed results for fairness metrics demographic parity, equalized odds, and equal opportunity.
>
> **Comment 1: Why do you use mathcal on a capital letter to denote a distribution?**
>
> We closely follow the same notation used in Madras et al. (2018) as we assume most of the readers would be familiar with their work, though we admit that notation could be improved.
>
> **Comment 2: There is $dz$ missing in Equation 4. Could you use f or g instead of $\mu$ there?**
>
> Thank you, we fixed the typo and added $dz$ in Equation 4. As before, we decided to follow the notation from Madras et al. (2018) and used $\mu$ to denote the optimal adversary, but we admit that this might have caused some confusion.
>
> **Comment 3: Could you be more explicit why you cannot carry out the change of variables for $p(z)$ for standard NN encoders?**
>
> Yes, we have now explicitly mentioned that the inverse function theorem cannot be applied.
>
> **Comment 4: Could you clarify the usage of Pinsker’s inequality as it bounds total variation and not MMD?**
>
> Yes, we now provide more details on this in Appendix A.1. As $\mu$ is a binary classifier, we can notice that $|\mu(z) \leq 1|$, so MMD/statistical distance is actually bounded by total variation, which is in turn bounded by Pinsker’s inequality. This was indeed not clear from the original text, and we hope our more detailed proof in Appendix A.1 should clarify it.
>
> **Comment 5: Can you elaborate usage of symmetrized KL divergence?**
>
> We have now also included more details on this in Appendix A.1. Essentially, we can notice that $TV(P, Q) = TV(Q, P)$, so we can apply Pinsker’s inequality twice to bound it using symmetrized KL divergence. We believe this is a minor detail of the algorithm, and other possible bounds could be used here.
>
> **Comment 6: The second instance of $\mu$  is a typo on the discussion of the optimal adversary at the end of page 1; it should be $\mu = 0$.**
>
> We could unfortunately not find this on Page 1. Could you please clarify where exactly this typo is in the paper?
>
> **Comment 7: Does Lemma 5.1 use normalizing flows?  How does the approach result in small adversarial accuracy? Could you provide some error bounds for your experiments?**
>
> Lemma 5.1 requires us to compute the optimal adversary $\mu^*$ which is based on comparing two densities $p_{Z_0}(z)$ and $p_{Z_1}(z)$. Here we use normalizing flows to compute these two densities exactly. While there could be some other approach, normalizing flows are currently the preferred solution for this type of problem. Note that the purpose of Lemma 5.1 is to estimate statistical distance *once training is finished*, and for the actual training of the encoder, we use Algorithm 1. Training is orthogonal, and we could use any training algorithm and still estimate statistical distance using Lemma 5.1 (naturally training should minimize statistical distance in some way, as otherwise statistical distance would be large).
> Finally, we did in fact show a 95% confidence bound (shaded region) computed using Lemma 5.1 in our experiment in Figure 5, where we showed that statistical distance indeed bounds adversarial accuracy in practice.

---

> > ### Comment · Reviewer_yCUe · 2021-11-16
> > **General response part 1**
> >
> > > Yes, we now provide more details on this in Appendix A.1.
> >
> > I like that you have added this section. Would it be possible to make the place where this is stated in the main (non appendix) a remark/prop/lemma, whilst simple this result is what directly justifies your algorithm thus its very important.
> >
> > > The second instance of  is a typo on the discussion of the optimal adversary at the end of page 1; it should be \mu=0
> >
> > Sorry it was page 5 not page 1 apologies must have missed this typo. It is page 5 paragraph second line bellow equation 4 second instance of \mu
> >
> > > Yes, Madras et al. (2018) have already proved that the statistical distance bounds various fairness metrics
> >
> > This makes sense however your theoretical results show the following:
> > - We can minimise a surrogate of statistical distance empirically justified via the TV/Pinkser bound
> > - once we have done this we can compute an estimtor of the statistical distance which concentrates with high probability around the true value.
> >
> > You dont gaurantee with with prob that the statistical distance will be low instead you gaurantee that you can provide a good (accurate/probable0 estimate of the statistical distance to the user (which hopefully is low). This should be made more clear in the abstract/intro as I felt if you give the paper an initial read you are claiming stronger results.  This implies that you should show the final statistical distance for all experiments.

---

> > ### Comment · Reviewer_yCUe · 2021-11-17
> > **Updated scores but please address the general response and update/or confirm the manuscript will be updated**
> >
> > I have updated my score to an 8. I think you have done an excellent job in adressing many of the highlighted issues and clarifying the confusion with \mu^*.  However I would like confirmation on the changes:
> >
> > 1. Changing the comment justifying the symmetrised KL as a surrogate to either a Remark, Lemma or Prop this needs enphasis as it is very important.
> > 2. The (...)dx=(...)dz manipulation. Whilst your result is correct It would be much nicer (and robust) if you could carry this out formally via a change of variables.
> > 3. The way you are selling the work in the intro/abstract in some way oversells the actual theoretical results. You have done good work but its important to be precise in the way you introduce it as being a paper in fariness there are high risks to not being precise. See general response for part 1.
> > 4. Correct the \mu=0 typo on page 5
> >
> > I need to do a final verification on the TV bound in apdx 1, but a quick skim does suggest it is fine. Great work !

---

> > > ### Author Response · Authors · 2021-11-18
> > > **Responses and updated manuscript**
> > >
> > > Thank you a lot for your positive and constructive feedback, and thank you for carefully checking our proofs. Your comments have really helped us to significantly improve the manuscript.
> > >
> > > We have now addressed your remaining points and updated the paper accordingly.
> > > Note that the numbering of lemmas and equations has changed slightly as we have now added two additional lemmas (one for definition of the optimal adversary and another for the bound with symmetrized KL) as you suggested. We provide our responses below.
> > >
> > > **Could you put your bound on statistical distance with symmetrized KL divergence as a separate lemma?**
> > >
> > > Yes, we have now put this bound as a separate lemma (now called Lemma 5.3), with the full proof in Appendix A.1.
> > >
> > > **Could you make the proof of the theorem with total variation in the Appendix (where you are saying (..)dx = (..)dz) more rigorous?**
> > >
> > > Yes, as you suggested, we have now written a more formal proof of the theorem by using the change of variables formula and manipulating Jacobians. Note that this proof is now on Page 16, and this theorem is now called Theorem 5.5 in the new revision.
> > >
> > > **Could you fix the typo in the second instance of the optimal adversary $\mu$ on Page 5?**
> > >
> > > Yes, we now understand what you meant, and we admit that the sentence was written in a slightly confusing manner. What we actually wanted to say there is that there are two different functions $\mu$ that attain the supremum, namely $\mu^*(z) = 1_{\{p_{Z_0}(z) \leq p_{Z_1}(z)\}}$ and $\mu^*(z) = 1_{\{p_{Z_1}(z) \leq p_{Z_0}(z)\}}$. As they both result in the same objective value, through the rest of the paper we assume, without loss of the generality, that $\mu^*(z) = 1_{\{p_{Z_0}(z) \leq p_{Z_1}(z)\}}$. We have now rewritten it to be more clear. Note that after the revision this part is now in Appendix A.1 (and there is separate Lemma 5.1 stating the form of an optimal adversary).
> > >
> > > **Could you describe the motivation for $\gamma$ weighting between the losses?**
> > >
> > > The motivation for this weighting comes from the fact that we are optimizing two (possibly competing) objectives: standard cross-entropy loss for classification $\mathcal{L}_{clf}$, and symmetrized KL loss to minimize the statistical distance (and, in turn, increase fairness) $\mathcal{L}_0 + \mathcal{L}_1$. This is an instance of a multi-objective optimization problem, as another reviewer has pointed out. In our case, we used convex scalarization, which is arguably the most common approach for this problem (see e.g. Madras et al.). In Appendix C, we experimented with another scalarization approach, namely Chebyshev scalarization from Wei & Niethammer (2020), and we obtained similar results. The tradeoff between accuracy and fairness is a well-known problem in fairness research, and our method could benefit from further advances in the area, as any scalarization technique can easily be plugged into our training algorithm.
> > >
> > > **Do you guarantee that you can provide a good (accurate/probable) estimate of statistical distance, and not that it will be low after training? Could you make it more clear in the abstract/intro?**
> > >
> > > Yes, this is precisely the correct understanding. Essentially, we have a training procedure in Algorithm 1 that we use to train encoders that, in practice, have a small statistical distance (as can be seen from experimental evaluation). Once training is finished, we can provide an estimate of the statistical distance that holds with high probability. Naturally, proving that training converges to the model with low statistical distance would be significantly more difficult (e.g., one would have to argue about the convergence of SGD for normalizing flows).
> > > Though we do not have that much space right now to elaborate in detail, we have slightly modified the introduction to make this point more clear (if paper is accepted we would have an extra page which we would use to clarify this further).

---

> ### Author Response · Authors · 2021-11-16
> **Response to Reviewer yCUe (Part 2)**
>
> Here we provide second part of our response:
>
> **Comment 8: Could you write Lemma 5.1 in the form $P(...) \geq 1 - \delta$?**
>
> Yes, thank you for the suggestion. We have now written Lemma 5.1 in this form to make it more straightforward for the reader.
>
>
> **Comment 9: Is the notation in the proof interchanging densities and measures? What is the purpose of calligraphy notation Z?**
>
> Here we again followed notation from Madras et al. (2018) so that readers familiar with that paper can more easily follow our derivations. So here $\mathcal{Z}_0$ and $\mathcal{Z}_1$ are two probability distributions, and $\mu$ is a binary classifier (not a measure).
>
> **Comment 10: Is there a misstep in the proof of Lemma 5.1?**
>
> Thank you for checking the proof in detail. You are undoubtedly correct that $\mu$ is undefined after removing the supremum. In fact, the idea of the proof is to replace the supremum with the evaluation at the optimal $\mu^*$ (that attains the supremum), introduced in Section 5. Now, we can apply our triangle inequality for this specific $\mu^*$ (otherwise, you are correct that with the supremum, one would have to assume the size of the hypothesis space and apply the union bound). We have now fixed this in the proof in the Appendix. Please let us know if you think there are still further issues.
>
>
> **Comment 11: Are flows a necessary part of this work? Do your guarantees also apply to standard encoders, while flows allow computing these statistical distances?**
>
> Yes, this is exactly right. Our proofs would apply to other encoders, but there would be no practical way to compute the statistical distance. Thus, without using normalizing flows, it would not be possible to perform any of the experiments in Section 6, so we believe flows are a crucial piece of the practical aspect of our work.
>
> **Comment 12: Could you push Jacobian determinant into the absolute value when carrying the change of variables?**
>
> We are not sure to which equation you are referring to here, but if you could point us to the exact place with the issue, we would be happy to correct this.
>
> **Comment 13: Does Lemma 5.2 motivate weighting with $\gamma$ in Algorithm 1?**
>
> Lemma 5.2 has a different purpose: for categorical datasets we show that we can explicitly construct a bijection that minimizes the statistical distance between the two distributions. This lemma is then used in the experiments with categorical datasets Compas and Adult to perform the encoding.

---

> > ### Comment · Reviewer_yCUe · 2021-11-16
> > **Union bound**
> >
> > Ok , I am going to reply prompty to see if we can address point 10 regarding the \sup. Heads up I still think you need to take a union bound, but lets try and go through the reasons step by step, like I said earlier even if this is the case I still believe some version of your theorem is correct and I am still optimistic there is time to fix this.
> >
> > Ultimately the statistical distance is a random quantity of the form  \sum_mu | R(mu) -R_hat(mu) | where the R's are risks and we are interested in bounding the probability P( \sup | R(mu) -R_hat(mu) |  > e)  (or its is compliment). How I am used to proceding here is that one derives a bound that is uniformly true for all \mu P( | R(mu) -R_hat(mu) |  > e)  < f(delta) and thus it will also be true for \mu*.
> >
> > Why is this ? well because I can only apply Hoeffding's inequality for a fixed \mu \in H , \mu^* is not quite a fixed \mu in H it is itself a random quantity whose distribution you have not characterised. In more succint terms you cannot apply Hoeffding to a \sup or an argsup (mu*) even if it is attained is an argmax. In other words u^* = argsup RV (of an RV) thus u^* (assuming it is attained for each omega) is itself a measurable map , a function of omega and a random variable.
> >
> > Take the following toy example (that motivates how u^* is a function of omega):
> >
> > say mu \in H = { x^2 , |x|}
> >
> > completely deterministic. Now say my  delta(mu) = ||\mu(0) - \mu(z) || where z ~ N(0,1)
> >
> > The quantity :
> >
> > \argmax_{\mu \in H} delta(mu)
> >
> > is an RV  simply because delta(mu) is an RV.  delta(mu) (omega) is a measurable map it depends on omega, if I fix omega (take a single sample from N(0,1) ) its a deterministic quantity however here its an rv
> > similarly the quantity:
> >
> > u^* = \argmax_{\mu \in H} delta(mu)
> >
> > is also a function of omega I get a different max potentially for every different value of omega. To see dependance on omega see how if samples lie in (-1,1) u^* = |x| meanwhile outside that range u^*=x^2.
> >
> >  Check example 2 here https://www.stat.purdue.edu/~jianzhan/STAT598Y/NOTES/slt04.pdf or these notes for example https://www.wisdom.weizmann.ac.il/~ethanf/teaching/ItSLT_16/lectures/lec2_no_anim.pdf .  In short you cant quite just plug in u^* as you do in the first step and Hoeffdings does not hold.
> >
> > I will look into the other responses (Pinksers ineq) tonight, btw its nice theres an Appendix 1 but I still think you should enphasis this point more in the paper, as you can see one of the reviewers missed it.

---

> > > ### Author Response · Authors · 2021-11-16
> > > **Response**
> > >
> > > Thanks a lot for your prompt reply! We believe that the main cause of the misunderstanding is that you are treating $\mu^*$ as a random variable, while for us it is a fixed function $\mu^*: R^d \rightarrow$ {$0, 1$} defined as $\mu^*(z) = 1_{\{p_{Z_0}(z) \leq p_{Z_1}(z)\}}$ (it evaluates to 1 if and only if $\{p_{Z_0}(z) \leq p_{Z_1}(z)\}$). This is a function because $p_{Z_0}$ and $p_{Z_1}$ are again fixed functions (not R.V.) which we can evaluate (recall that this is exactly why we need normalizing flows). We can then proceed by applying Hoeffding’s inequality as in Appendix. In our paper $\mu^*$ is defined as a function right after Equation 5, but if you think we could make it more clear, we would be happy to add more details in the paper.
> > >
> > > Conceptually, we are interested in bounding the error for a single fixed optimal classifier $\mu^*$, which in our setup we can compute exactly, and we do not need to estimate the worst case error of any classifier $\mu$ (there your approach would certainly be necessary as we would have to ensure that the inequality holds for all possible $\mu$ simultaneously by taking union bound over the samples in the entire hypothesis space, as you suggested). But, in our work this is not necessary as we are using normalizing flows to compute $p_{Z_0}(z)$ and $p_{Z_1}(z)$ exactly.

---

> > > > ### Comment · Reviewer_yCUe · 2021-11-16
> > > > **see the toy example, mu is deterministic yet mu^* is not**
> > > >
> > > > Its not valid to treat mu^* as a deterministic function when it is the result of taking a sup of a random quantity, if we could do this PAC bounds would be completely trivial (and also not particularly useful). What you are telling me is that mu* is not the result of taking the sup ? or you are saying that the result of taking this sup is some how deterministic ? (it sounds like you are claiming the second and you do seem to have a proof for this but you need to help me out and connect the statements more carefully because I am missing the full story)
> > > >
> > > > If we go back to my example \mu \in H = {x^2, |x|} you can see that  mu^* is a random variable despite candidate \mu's begin deterministic functions. I cant just go ahead and decide I will treat \mu^* as a deterministic quantity because I want to when its the result of  stochastic quantity .
> > > >
> > > > Again to enphasize you cant treat mu^* as a fixed function  when your statistical distance is itself a random quantitiy (it is , the whole point of this theorem is that you are bounding probability of the the sup of this random quantity if it wasnt random you wouldnt be doing concetration bounds in probability ).
> > > >
> > > > Please revisit the toy example I provided that illustrates why you cant treat \mu^* as a deterministic quantity. I follow how the flows allow you to in practice estimate a fixed mu for a given fixed \omega (sample) and I do think that is fine, however the bounds you are doing are in prob and over all omega it is very much a classical PAC bound and I am not convinced you can just adhocly treat \mu^* as deterministic here.
> > > >
> > > > Back to my understanding of your claim: you are saying that \mu has to be deterministic ? that is when you take the argsup the pushforward induced collapses to a delta. If thats the case you need to formally prove this, I dont see how stating it is enough . I do agree that the confusion is that I am seeing \mu^* as coming out of an argsup whilst you are formally defining it in terms of this indicator function (which you seem to prove minimises this argsup, you need to cite back to this proof in 5.1 connect the statements carefully etc). I am moving towards understanding it better but still not completely convinced as I am going back and forth around different statemetns that are still not well connected (referencing each other) in the paper  .
> > > >
> > > > In short:
> > > > 1. If mu comes out of the argsup of an RV then mu would typically be an RV. Can you prove it does not depend on \omega ?
> > > > 2. You have a construction of \mu in terms of an indicator and you argue that this is in fact the sup over the random statistical distance ? are you saying a random quantity has a determnistic mnimiser (this is possible but it needs proof, and if the proof is already there I am missing how the statements connect etc).

---

> > > > > ### Comment · Reviewer_yCUe · 2021-11-16
> > > > > **recap**
> > > > >
> > > > > Ok I think I see it. Here are some comments:
> > > > >
> > > > > 1. You show that Eq 4 is minimised by a single deterministic u, so the argsup of the statistical distance is not random
> > > > > 2. In the appendix please recap this when you are taking the step of plugging the argsup into the statistical distance
> > > > > 3. I would recommend a clear (with assumptions) Lemma or a remark for \mu* = indicator ... rather than a pargraph and you should ref this in the proof of 5.1
> > > > >
> > > > > Here are some questions/pints on increasing clarity:
> > > > >
> > > > > 1. Why are you using \Delta(\calZ_1, \calZ_2)  in the appendix and instead \Delta(p_Z1, p_Z2) in eq for in the main ? I find this confusing because you derive (prove) the unique minimiser \mu* using the p_Z notation but then in the appendix you are using the \calZ notation and plugin in the minimiser \mu of  \Delta(p_Z1, p_Z2) . So does  \Delta(p_Z1, p_Z2)  = \Delta(\calZ_1, \calZ_2) , if so please adhere to one notation, if they are not the same then why would the defined \mu* of one work for the other?
> > > > > 2. Finally if \mu* is not random I would like some intution as to what \Delta(\calZ_1, \calZ_2)  is, as in is this quantity random or not ? based on the bounds you are computing you are treating the quantity as random, yet somehow the argsup is not, a bit more detail into this would be helpful. If in fact \Delta(\calZ_1, \calZ_2)  is not random then more detail needs to be put into describing what is  (is it the empirical subsamples only that are providing the randomness).
> > > > >
> > > > > It is strange the different treatment given here than what is classically done in PAC and I am just trying to iron out the differences that allow your choices to be valid and make sure they are correct and clear.

---

> > > > > > ### Comment · Reviewer_yCUe · 2021-11-16
> > > > > > **recap^2 (I think I am convinced now, but make things more clear please)**
> > > > > >
> > > > > > Note the key difference seems to be that the statistical distance which has the sup is in fact deterministic whilst the random quantity is the empirical statistical distance , and you are not taking a \sup of these two as done in PAC so in fact the \sup considered at the start is over a deterministic quantity, I think this could be more clear just see some of the suggestions above , it might seem pedantic but this is a fairness paper and it will be read by people with a wide variery of backgrounds looking for some notionof "safety" so you should be extremely clear.

---

> > ### Comment · Reviewer_yCUe · 2021-11-16
> > **General response  for part 2**
> >
> > > We are not sure to which equation you are referring to here, but if you could point us to the exact place with the issue, we would be happy to correct this. (comment 12)
> >
> > Proof of theorem 5.3 in the appendix  you are saying (..)dx = (..)dz final paragraph of the proof on page 15. Read the original comment again with this in mind. Heuristics with infinetisimals are handwavy, whilst the result is correct I would be much happier if you used the full change of variables formula (via the inverse transform theorem)  and manipulate the jacobians that arise properly to show the two TVs are the same, the dx=dv argument is handwavy and there are cases where it could fail like in the example I gave.
> >
> > > Lemma 5.2 has a different purpose:
> >
> > Could you briefly describe if the gamma weigting you use in the algorithm has a formal motivation, its fine if its just a heuristic but I think the motivation could be enphasized more.
> >
> > Otherwise I am quite happy with part 2 (subject to the clarifications we discussed in the other thread).

---

### Official Review · Reviewer_xsbe · 2021-11-03

**Correctness:** 3
**Technical Novelty And Significance:** 3
**Empirical Novelty And Significance:** 3
**Recommendation:** 6
**Confidence:** 3

**Main Review:**

[Strengths]
- Normalizing flows are a flexible tool to disentangle the predictive information and sensitive attributes from the data.
- The idea of using normalizing flows for fair representation learning is interesting. The formulation of the learning problem is natural and practical.
- The presentation of the paper is clear.

[Weaknesses]
- The proposed method applies to only a binary protected attribute. To generalize to a discrete protected attribute, we would need to learn a separate flow for each value. However, equalizing all these distributions is not straightforward as the symmetric KL can no longer be used.
- I think the proposed method can sacrifice accuracy for fairness. Suppose that the label is a function of the input distribution, normalizing the input distribution could also remove the predictive information. The authors use a parameter \gamma in the loss function to balance fairness and classifier utility; however, this is not sufficient to attain a good tradeoff in terms of Pareto optimality.
- In the empirical evaluation, the authors used different density functions, i.e., RealNVP, MADE, and GMMs, for different datasets. This led me wonder if some of the densities do not perform well on certain datasets. For a fair comparison, it could be helpful to provide results for all the densities.


**Summary Of The Paper:**

In this paper, the authors propose fair representation learning using normalizing flows. The basic idea is to learn a separate normalizing flow for each protected group. These flow distributions are equalized and hence cannot be used to determine the protected group. The authors argue that improved fairness guarantees can be achieved compared to adversarially fair learning.

**Summary Of The Review:**

The idea of the paper is clear and interesting. It appears that the proposed approach only supports a binary protected attribute. The empirical evaluation could be strengthened by providing results for all the density functions.

---

> ### Author Response · Authors · 2021-11-16
> **Response to Reviewer xsbe**
>
> Thank you for taking the time to provide this insightful review. We address your comments below.
>
> **Does your method sacrifice accuracy for fairness? Did you investigate the Pareto front of your method?**
>
> Yes, we investigated different multi-objective optimization techniques in Appendix C, where we showed that the convex scalarization and the Chebyshev scalarization scheme achieve similar performance in terms of fairness-accuracy tradeoff. Accordingly, we believe that the Pareto front of our method is convex, as the convex scalarization scheme typically only recovers points on the convex hull of the Pareto front. Moreover, we also showed that our method achieves roughly the same accuracy as prior fair representation learning approaches while being the first to provide provable guarantees on the maximum adversarial accuracy.
>
>
> **Why do you use different density function estimates for different datasets?**
>
> As detailed in Section 6, we estimated the input densities with RealNVP for Health, MADE for Adult and Compas, and Gaussian Mixture Models for Law School and Crime. Since Adult and Compas are discrete datasets, we cannot use the same density estimation as for the other datasets. We use RealNVP for Health as it is a more complex dataset. Furthermore, in Appendix C, we demonstrated that our method is compatible with any differentiable density estimate, as it achieves the same performance with Gaussian Mixture Models, an autoregressive prior, and RealNVP.
>
>
> **Does the current formulation of your method only work for binary protected attributes?**
>
> As you mention, the extension to categorical protected attributes would require equalizing multiple normalizing flows, which is not straightforward. However, we consider this an exciting direction for future work.

---

> > ### Comment · Reviewer_xsbe · 2021-11-29
> > **updated the score**
> >
> > Thanks for answering the questions. I've increased the score accordingly.

---

### Official Review · Reviewer_My3D · 2021-11-03

**Correctness:** 3
**Technical Novelty And Significance:** 3
**Empirical Novelty And Significance:** 2
**Recommendation:** 6
**Confidence:** 2

**Main Review:**

In general, I think the paper is well presented and the organization is good. My major concern is the motivation of this paper, I would like to see why normalization flow is the necessary technique to be the solution to the drawbacks mentioned in existing fair representation learning methods. In addition, the theory seems to be quite straightforward based on PAC learning. If more tailored theoretical analysis can be provided, I think the paper can be more interesting. While I have the above concerns, the technique part in this paper is sound, and I cannot find large mistakes, thus I vote for a weak acceptance.

**Summary Of The Paper:**

This paper aims to design a fair representation learning method. To achieve this goal, the authors have proposed a normalization flow-based method, where an adversarial-based strategy is leveraged to train the model. In the experiments, the authors conducted extensive experiments to demonstrate the effectiveness of the proposed methods.

**Summary Of The Review:**

Interesting paper, which however can be further improved.

---

> ### Author Response · Authors · 2021-11-16
> **Response to Reviewer My3D**
>
> Thank you for your encouraging words and positive feedback, which we address below.
>
> **Why are normalizing flows necessary to solve the drawbacks of prior fair representation learning methods?**
>
> As mentioned in Section 5, for standard models, given densities $p_0(x)$ and $p_1(x)$ over the input space, it is intractable to compute the densities $p_{Z_0}(z)$ and $p_{Z_1}(z)$ in the latent space as many inputs $x$ can be mapped to the same latent $z$ and we thus cannot use the inverse function theorem. Consequently, prior adversarial training methods cannot compute the optimal adversary and thus resort to a lower bound.

---

### Author Response · Authors · 2021-11-16
**General response**

We thank the reviewers for their insightful and positive feedback. We are encouraged that they found our paper well-presented (`R-My3D`, `R-xsbe`) and well-motivated (`R-yCUe`, `R-Vaqy`) and that they considered our idea of using normalizing flows for fair representation learning with theoretical guarantees significant, promising, and practical (`R-xsbe`, `R-yCUe`, `R-BBgE`). Below, we individually address the reviewers’ comments. We have also incorporated feedback in the new revision.

---

### Author Response · Authors · 2021-11-26
**Final questions**

Dear reviewers,

As the end of the discussion is approaching, we would like to kindly ask you to consider our responses to your concerns.
We answered each of your questions, and updated the manuscript based on your feedback.
Thank you for engaging into the discussion so far, and please let us know if you have any final questions that we can address.

Authors

---

### Decision · Program_Chairs · 2022-01-20

**Decision:**

Accept (Poster)

**Comment:**

This paper addresses fair representation learning, with the aim of obstructing the recovery of sensitive features from the learned representation, hence enforcing the fairness of subsequent prediction tasks.  In the setting where probability density can be estimated for sensitive groups, Fair Normalizing Flows (FNF) tries to minimize the statistical distance between group-wise latent representations, thereby providing theoretical fairness guarantees.  Experimental confirm the effectiveness of FNF in fairness, transferrability, and interpretability.

The paper received extensive and in-depth discussion.  The rebuttal did an excellent job in clarification.  Although there are still some concerns on the theoretical properties of the optimal solution, overall the reviewers and myself find this paper interesting and worth publishing.